

# 1 Perfluorocyclobutane (PFC-318, $c$-C$_4$F$_8$) in the global
# 2 atmosphere

Jens Mühle[1], Cathy M. Trudinger[2], Matthew Rigby[3], Luke M. Western[3], Martin K. Vollmer[4],
Sunyoung Park[5], Alistair J. Manning[6], Daniel Say[3], Anita Ganesan[7], L. Paul Steele[2], Diane J.
Ivy[8], Tim Arnold[9,10], Shanlan Li[5], Andreas Stohl[11], Christina M. Harth[1], Peter K. Salameh[1],
Archie McCulloch[3], Simon O'Doherty[3], Mi-Kyung Park[5], Chun Ok Jo[5], Dickon Young[3], Kieran
M. Stanley[3], Paul B. Krummel[2], Blagoj Mitrevski[2], Ove Hermansen[11], Chris Lunder[11], Nikolaos
Evangeliou[11], Bo Yao[12], Jooil Kim[1], Benjamin Hmiel[13], Christo Buizert[14], Vasilii V. Petrenko[13],
Jgor Arduini[15,16], Michela Maione[15,16], David M. Etheridge[2], Eleni Michalopoulou[3], Mike
Czerniak[17], Jeffrey P. Severinghaus[1], Stefan Reimann[4], Peter G. Simmonds[3], Paul J. Fraser[2],
Ronald G. Prinn[8], and Ray F. Weiss[1]
[1]Scripps Institution of Oceanography, University of California, San Diego, La Jolla, CA, USA
[2]Climate Science Centre, CSIRO Oceans and Atmosphere, Aspendale, Victoria, Australia
[3]School of Chemistry, University of Bristol, Bristol, UK
[4]Laboratory for Air Pollution and Environmental Technology, Empa, Swiss Federal Laboratories for Materials
Science and Technology, Dübendorf, Switzerland
[5]KNU, Kyungpook Institute of Oceanography, College of Natural Sciences, Kyungpook National University, South
Korea
[6]Met Office Hadley Centre, Exeter, UK
[7]School of Geographical Sciences, University of Bristol, Bristol, UK
[8]Center for Global Change Science, Massachusetts Institute of Technology, Cambridge, MA, USA
[9]National Physical Laboratory, Teddington, Middlesex, UK
[10]School of GeoSciences, University of Edinburgh, Edinburgh, UK
[11]NILU, Norwegian Institute for Air Research, Kjeller, Norway
[12]Meteorological Observation Centre (MOC), China Meteorological Administration (CMA), Beijing, China
[13]Department of Earth & Environmental Sciences, University of Rochester, Rochester, NY, USA
[14]College of College of Earth, Ocean, and Atmospheric Sciences, Oregon State University, Corvallis, OR, USA
[15]Department of Pure and Applied Sciences, University of Urbino, Urbino, Italy
[16]ISAC-CNR, Bologna, Italy
[17]Edwards LTD, Burgess Hill, West Sussex, UK

*Correspondence to*: Jens Mühle (jmuhle@ucsd.edu)



**Abstract.** We reconstruct atmospheric abundances of the potent greenhouse gas $c$-$C_4F_8$ (perfluorocyclobutane,
perfluorocarbon PFC-318) from measurements of in situ, archived, firn, and aircraft air samples with precisions of
~1–2 % reported on the SIO-14 gravimetric calibration scale. Combined with inverse methods, we found near zero
atmospheric abundances from the early 1900s to the early 1960s, after which they rose sharply, reaching 1.66 ppt
(parts per trillion dry-air mole fraction) in 2017. Global $c$-$C_4F_8$ emissions rose from near zero in the 1960s to ~1.2 Gg
$yr^{-1}$ in the late 1970s to late 1980s, then declined to ~0.8 Gg $yr^{-1}$ in the mid-1990s to early 2000s, followed by a rise
since the early 2000s to ~2.2 Gg $yr^{-1}$ in 2017. These emissions are significantly larger than inventory based emission
estimates. Estimated emissions from eastern Asia rose from 0.36 Gg $yr^{-1}$ in 2010 to 0.73 Gg $yr^{-1}$ in 2016 and 2017,
31 % of global emissions, mostly from eastern China. We estimate emissions of 0.14 Gg $yr^{-1}$ from Northern and
Central India in 2016 and find evidence for significant emissions from Russia. In contrast, recent emissions from
North Western Europe and Australia are estimated to be small ($\leq$ 1 % each). We conclude that emissions from China,
India and Russia are likely related to production of polytetrafluoroethylene (PTFE, "Teflon") and other
fluoropolymers that are based on the pyrolysis of hydrochlorofluorocarbon HCFC-22 ($CHClF_2$) in which $c$-$C_4F_8$ is a
known by-product. The semiconductor sector, where $c$-$C_4F_8$ is used, is estimated to be a small source. Without an
obvious correlation with population density, incineration of waste containing fluoropolymers is probably a minor
source, and we find no evidence of emissions from electrolytic production of aluminum in Australia. While many
possible emissive uses of $c$-$C_4F_8$ are known, the start of significant emissions may well be related to the advent of
commercial PTFE production in 1947. Process controls or abatement to reduce $c$-$C_4F_8$ by-product were probably not
in place in the early decades, explaining the increase in emissions. With the advent of by-product reporting
requirements to the United Nations Framework Convention on Climate Change (UNFCCC) in the 1990s, concern
about climate change and product stewardship, abatement, and perhaps the collection of $c$-$C_4F_8$ by-product for use in
the semiconductor industry where it can be easily abated, it is conceivable that emissions in developed countries
were stabilized and then reduced, explaining the observed emission reduction in the 1980s and 1990s. Concurrently,
production of PTFE in China began to increase rapidly. Without emission reduction requirements, it is plausible that
global emissions today are dominated by China and other developing countries, in agreement with our analysis. We
predict that $c$-$C_4F_8$ emissions will continue to rise and that $c$-$C_4F_8$ will become the second most important emitted
PFC in terms of $CO_2$-equivalent emissions within a year or two. The 2017 radiative forcing of $c$-$C_4F_8$ (0.52 mW $m^{-2}$)
is small but emissions of $c$-$C_4F_8$ and other PFCs, due to their very long atmospheric lifetimes, essentially
permanently alter Earth's radiative budget and should be reduced. Significant emissions outside of the investigated
regions clearly show that observational capabilities and reporting requirements need to be improved to understand
global and country scale emissions of PFCs and other synthetic greenhouse gases and ozone depleting substances.




## 1 Introduction


The perfluorocarbon (PFC) perfluorocyclobutane ($c$-$C_4F_8$, PFC-318, octafluorocyclobutane, CAS 115-25-3) is a very
long-lived and potent greenhouse gas (GHG) regulated under the Paris Agreement of the United Nations Framework
Convention on Climate Change (UNFCCC). Ravishankara et al. (1993) concluded that the most important
atmospheric loss process of $c$-$C_4F_8$ is Lyman-α photolysis resulting in an atmospheric lifetime of 3200 years. Later,
Morris et al. (1995) argued that if reactions of $c$-$C_4F_8$ with electrons and positive ions in the mesosphere and aloft are
irreversible, the lifetime could be reduced to 1400 years, which, on human timescales, is still essentially infinite. $c$-
$C_4F_8$ has a radiative efficiency of 0.32 W $m^{-1}$ $ppb^{-1}$ (parts per billion) and, assuming a 3,200 year lifetime, a global
warming potential of 9,540 on a 100-year timescale ($GWP_{100}$) (Myhre and Shindell et al., 2013; Engel and Rigby et
al., 2018). Due to the long lifetime and high radiative efficiency, emissions of $c$-$C_4F_8$ (and other perfluorinated
compounds) essentially permanently alter the radiative budget of Earth (Victor and MacDonald, 1999).
Lovelock (1971) predicted the accumulation of $c$-$C_4F_8$ in the global atmosphere, but to the best of our knowledge, the
earliest atmospheric measurements of $c$-$C_4F_8$ were presented in Sturges et al. (1995) and in the Ph.D. theses of
Travnicek (1998) and Oram (1999, discussed further below). Sturges et al. (2000) determined from one vertical
balloon-borne profile in 1994 that $c$-$C_4F_8$ mole fractions declined from ~1.1 ppt (parts per trillion) in the lower
atmosphere of the Northern Hemisphere (NH) to ~0.6 ppt in the stratosphere, while Harnisch (1999) reported that
Sturges et al. (1995) had found 0.4 ppt in the troposphere decreasing to ~0.1 ppt at 25 km in 1994, suggesting a
revised calibration scale. Harnisch et al. (1998; 1999) estimated from this atmospheric gradient global emissions of
1–2 Gg $yr^{-1}$ (kt $yr^{-1}$). Travnicek (1998) reported ~0.2 ppt in 1977 and ~0.7 ppt in 1997 in the NH troposphere, from
which Harnisch (2000) estimated average global emissions of 0.7 Gg $yr^{-1}$. Despite differences in early measurements
and emissions estimates, perhaps due to different calibration scales and analytical methods, these studies were
consistent with the accumulation of $c$-$C_4F_8$ in the global atmosphere.
Harnisch (1999, 2000) stated that $c$-$C_4F_8$ had limited economic relevance, with some use for plasma etching in the
semiconductor industry, that $c$-$C_4F_8$ can be formed via dimerization of tetrafluoroethylene (TFE), and that thermal
decomposition or combustion of polytetrafluoroethylene (PTFE) and other fluoropolymers (Morisaki, 1978) (during
waste disposal) possibly led to the accumulation of atmospheric $c$-$C_4F_8$.
Today we have stronger evidence for $c$-$C_4F_8$ emissions from the semiconductor and microelectronics industry as it
has been increasingly used since the 1990s for dry etching, chemical vapor deposition chamber cleaning and as
deposition gas (Bosch process). Compared to other fluorinated gases used for these processes, more selective
etching, cost reduction in plasma cleaning, easier abatement and hence potentially lower contribution to global
warming have been cited as advantages of $c$-$C_4F_8$ (e.g., Sasaki et al., 1998; Christophorou and Olthoff, 2001; Raju et
al., 2003; Kokkoris et al., 2008; and references therein). However, due to efficient abatement with modern emission
controls (up to 90 %), today's $c$-$C_4F_8$ emissions from this industry could also be small (Zhihong et al., 2001).
Recently there is also further evidence that the thermal decomposition of PTFE and other fluoropolymers can lead to
the formation of $c$-$C_4F_8$, TFE and hexafluoropropylene (HFP) (van der Walt et al., 2008; Bezuidenhoudt et al., 2017);
the resultant $c$-$C_4F_8$ could therefore be emitted to the atmosphere.





One potentially major source of $c$-C$_4$F$_8$ that seems to have received too little attention, is the production of TFE and
HFP monomers, the building blocks for PTFE, fluorinated ethylene propylene (FEP, TFE/HFP copolymer) and other
fluoropolymers, which involves pyrolysis of hydrochlorofluorocarbon 22 (HCFC-22, CHClF$_2$) as $c$-C$_4$F$_8$, the dimer
of TFE, is a by-product/intermediate of this process (Chinoy and Sunavala, 1987; Broyer et al., 1988; Gangal and
Brothers, 2015). This reaction can be steered towards HFP or $c$-C$_4$F$_8$ by controlling the dimerization of TFE to $c$-
C$_4$F$_8$ and the co-pyrolysis of $c$-C$_4$F$_8$ with TFE to HFP (Jianming, 2006). $c$-C$_4$F$_8$ could therefore be emitted during
TFE/HFP/PTFE/FEP production if it is not abated or recovered, e.g. for use in the semiconductor industry or for
pyrolysis with TFE to HFP at a later stage, perhaps at a different facility.
Several other, perhaps minor, emissive uses of $c$-C$_4$F$_8$ are also known (see Lewis, 1989; Chung and Bai, 2000;
Harnisch, 2000; Christophorou and Olthoff, 2001; Kim et al., 2002; Liu et al., 2008; and reference therein), e.g., in
aerolyzed foods, as a food packaging gas, in retinal detachment surgery, for contrast-enhanced ultrasound imaging,
in radar systems, as a specialty refrigerant (e.g., in submarines where R405A (43 % $c$-C$_4$F$_8$) can replace pure HCFC-
22 and chlorofluorocarbon CFC-12), as an electrically insulating dielectric gas (e.g., in mixtures with sulfur
hexafluoride, SF$_6$), as a medium for polymerization reactions, in fire extinguishers, to estimate the size of natural gas
and oil reservoirs, for leak detection of nuclear waste containers (Schmidbauer, N., personal communication, 2011)
and as a geohydrological tracer (Kass, 1998). Production of $c$-C$_4$F$_8$ for these uses, via the pyrolysis of HCFC-22 or
perhaps from 1,2-dichlorotetrafluoroethane (CFC-114) (Siegemund et al., 2016), may cause emissions as well. While
the two major atmospheric PFCs, tetrafluoromethane (CF$_4$) and hexafluoroethane (C$_2$F$_6$), and the minor PFC octa-
fluoropropane (C$_3$F$_8$) are released during primary aluminum production (Holliday and Henry, 1959; Tabereaux,
1994; Fraser et al., 2013), no evidence for $c$-C$_4$F$_8$ emissions from primary aluminum production has been presented
so far. Cai et al. (2018) presented evidence for negligible emissions of $c$-C$_4$F$_8$ from the similar electrolytic production
of rare earth elements in China. There are no known natural sources of $c$-C$_4$F$_8$. In summary, there may be multiple $c$-
C$_4$F$_8$ emission sources, but the extent and time evolutions of these various potential emission sources are unclear.
Saito et al. (2010) reported the first continuous, approximately four year long, in situ measurement record of $c$-C$_4$F$_8$
at two stations in the NH, with mean baseline 2006–2009 mole fractions of ~1.22 ppt at Cape Oshiishi (43.1° N,
145.3° E) and ~1.33 ppt at Hateruma Island (24.1° N, 123.8° E) (NIES calibration scale). Saito et al. (2010)
determined increase rates of 0.01–0.02 ppt yr$^{-1}$ and global emissions of 0.6 ± 0.2 Gg yr$^{-1}$.
Oram et al. (2012) published the first multi-decade long atmospheric record of $c$-C$_4$F$_8$ in the Southern Hemisphere
(SH). They combined previous measurements of sub-samples of the Cape Grim Air Archive (CGAA) for the SH
with air dates prior to 1994 (from Oram, 1999, converted to a new, 19.6 % lower calibration scale with an estimated
uncertainty of ≤7 %) with newer measurements of CGAA sub-samples with air dates after 1994 and a change of
analytical method after 2006. They found an increase of $c$-C$_4$F$_8$ at Cape Grim from 0.35 ppt in 1978 to ~0.8 ppt in
1995 and 1.2 ppt in 2010, with a current increase rate of ~0.03 ppt yr$^{-1}$. They reported that global $c$-C$_4$F$_8$ emissions
increased from ~0.9 Gg yr$^{-1}$ in the early 1980s to ~1.7 Gg yr$^{-1}$ in 1986 before declining to a minimum of ~0.4 Gg yr$^{-1}$
in 1993, after which they increased to ~1.1 Gg yr$^{-1}$ in 2006 and 2007 and may have stabilized. Oram et al. (2012)
noted that the global emissions determined by Saito et al. (2010) were lower than their estimate and suggested that
the underlying atmospheric rise rate measured by Saito et al. may be too small.





In summary, calibration differences between previous studies are significant, no multi-decadal $c$-C$_4$F$_8$ record for the
NH has been published, and global emissions have not been reassessed since Oram et al. (2012). Therefore our
primary goals have been to develop an independent gravimetric $c$-C$_4$F$_8$ calibration scale and to characterize the
abundance of $c$-C$_4$F$_8$ with high precisions in both hemispheres in order to determine updated historic and recent
global emissions. We present measurements of $c$-C$_4$F$_8$ with precisions of ~1–2 % on the SIO-14 calibration scale (~2
% accuracy) developed by the Scripps Institution of Oceanography (SIO) using instrumentation and calibration
methods of the Advanced Global Atmospheric Gases Experiment (AGAGE) program (Prinn et al., 2018). We
discuss historic atmospheric mole fractions of $c$-C$_4$F$_8$ based on measurements of the CGAA for the extra-tropical SH,
archived air samples from various sources for the extra-tropical NH, continuous atmospheric measurements in both
hemispheres at multiple remote AGAGE stations since mid-2010, combined with measurements of air extracted from
firn from both hemispheres. Using our measurements and inverse modelling methods, we infer global $c$-C$_4$F$_8$
emissions since the beginning of the 20$^{th}$ century until 2017. To improve our understanding of prominent $c$-C$_4$F$_8$
sources and source regions, we investigate regional $c$-C$_4$F$_8$ emission strengths as observed by the global AGAGE
network in eastern Asia, Europe, parts of Australia and Russia and by an aircraft campaign over India. We also
summarize and discuss available inventory based "bottom-up" emissions and compare them to the emissions we
determined with our atmospheric measurement based "top-down" approach.
**2 Experimental methods**
**2.1 Instrumentation, data availability, and calibration**
$c$-C$_4$F$_8$ and ~40 other halogenated compounds were measured by AGAGE in 2 L air samples with the "Medusa"
cryogenic pre-concentration system with gas chromatograph (GC, Agilent 6890) and quadrupole mass selective
detector (MSD) (Miller et al., 2008; Prinn et al., 2018). Data from twelve in situ measurements sites and fourteen
Medusa instruments were used. At Monte Cimone, Italy, $c$-C$_4$F$_8$ was measured with a commercial Adsorption-
Desorption System with gas chromatograph and mass spectrometer (ADS-GC/MS) (Maione et al., 2013). Table 1
shows the availability of in situ, archived air (Sect. 2.2), firn air (Sect. 2.3) and aircraft air sample (Sect. 2.4)
measurements with information for each site. For all measurements, each sample was alternated with a reference gas
(Prinn et al., 2000; Miller et al., 2008), resulting in up to 12 fully calibrated samples per day (Medusa and ADS-
GC/MS). The reference gases at each site were calibrated relative to parent standards at SIO.
$c$-C$_4$F$_8$ measurements are reported on the SIO-14 calibration scale as ppt dry-air mole fractions. The calibration scale
is based on four gravimetric halocarbon/nitrous oxide (N$_2$O) mixtures via a stepwise dilution technique with large
dilution factors for each step ($10^3$ to $10^5$) (Prinn et al., 2000; 2001). High purity $c$-C$_4$F$_8$ (99.999 %, Matheson Trigas)
and N$_2$O (99.9997 %, Scott Specialty Gases) were further purified by repeated cycles of freezing (-196° C), vacuum
removal of non-condensable gases, and thawing. Artificial air (Ultra Zero Grade, Airgas) was further purified via an
absorbent trap filled with glass beads, Molecular Sieve (MS) 13X, charcoal, MS 5Å, and Carboxen 1000 at -80° C
(ethanol/dry ice). Zero air was measured to verify insignificant $c$-C$_4$F$_8$ and other halocarbon blank levels before
being spiked with the $c$-C$_4$F$_8$/N$_2$O mixtures. The resulting mixtures of $c$-C$_4$F$_8$ in artificial air have prepared values of



~1.3 ppt and the relative standard deviation of the calibration scale is 0.23 %. We estimate the uncertainties of the
calibration scale propagation from SIO to the sites to be ~0.6 % and the calibration scale uncertainty to be ~2 %.
The primary calibration instrument for the AGAGE network at SIO (La Jolla, California), Medusa 1, and all field
instruments used a Porabond Q (25 m, 0.32 mm I.D., 5 µm film thickness, Varian) chromatographic main column
and, initially Agilent 5973, later 5975 series MSDs. The original Medusa design is described by Miller et al. (2008);
subsequently all Medusas were converted or newly built to measure nitrogen trifluoride ($NF_3$) (Arnold et al., 2012),
but this did not affect the $c$-$C_4F_8$ measurements methodology or the results. While 5975 MSDs are beneficial for
samples and compounds with very low mole fractions, precisions for $c$-$C_4F_8$ measurements of archived air samples
(3–7 replicates, see next section) were similar, i.e. better than ~0.01 ppt. Daily reference gas measurement precisions
slightly improved from ~0.02 ppt (~1.5–2 %) to ~0.01 ppt (~1–1.5 %) with the 5975 MSDs. Detection limits (3 times
baseline noise) for 2 L air samples were ~0.01–0.03 ppt, perhaps slightly better for 5975 MSDs.
In addition to calibrations, Medusa 1 was also used to measure in situ local ambient air and several archived air
samples (see Sect. 2.2). However, analysis of most archived air samples at SIO occurred on a second instrument,
Medusa 7, as it was equipped with a more sensitive 5975 MSD at that time. For these measurements, we temporarily
converted Medusa 7 to use a GasPro GSC (60 m, 0.32 mm I.D., Agilent) main column as it promised better
separation performance for several higher PFCs (Ivy et al., 2012) measured along with $c$-$C_4F_8$. Similarly, Medusa 9,
the instrument used to measure most CGAA samples at the Commonwealth Scientific and Industrial Research
Organisation (CSIRO, Aspendale) and ambient air after October 2010, had been converted to use a GasPro column.
On both types of main columns, $c$-$C_4F_8$ was measured on mass over charge ratios (m/z) of 131 ($C_3F_5^+$) and 100
($C_2F_4^+$) and reported by height using carefully chosen integration parameters as perfluorobutane ($C_4F_{10}$) shares both
m/z and elutes on the tail of $c$-$C_4F_8$. The m/z ratios remained the same despite the very different separation principles
of these two main columns. Measurements of archived air samples on Medusa 7 with both main columns agreed
within less than 0.01 ppt (ratio of 1.0016, $R^2$ = 1.0000, n = 4, 0.237–1.11 ppt). In situ $c$-$C_4F_8$ measurements at SIO
with Medusa 1 (Porabond Q) and 7 (with the GasPro column) continued to agree within typical precisions. We also
compared archived air measurements on Medusa 1 and 7, both before and while Medusa 7 used the GasPro column,
and results agree within precisions of 0.02 ppt or better (Medusa 1 vs. Medusa 7, both Porabond Q, ratio of 1.0001,
$R^2$ = 0.9987, n = 95, 0.237–1.616 ppt, Medusa 1, Porabond Q vs. Medusa 7, GasPro, ratio of 1.0018, $R^2$ = 0.9979, n
= 39, 0.239–1.515 ppt). These tests show that the different main columns did not cause any bias.
The analytical systems showed no significant $c$-$C_4F_8$ blanks. The linearity of Medusa 7 (SIO) and 9 (CSIRO) used to
measure archived air samples was assessed with a series of diluted air samples (parent tank at 1.252 ppt, dilutions
from 100 % to 6.25 %, Ivy et al., 2012) and a series of different volumes of a working standard (parent tank at 1.60
ppt, sample volumes from 200 % to 5 % of usual 2 L volume). A small deviation from linearity was observed for the
most diluted samples and the smallest volumes probably due to a memory or blank of ~0.014 ppt on Medusa 9 was
corrected for. Medusa 7 showed perhaps an effect of ~0.008 ppt, but as this was just below the detection limits and
within the typical precisions, we chose not to correct for this.





**2.2 Archived air samples of the extra-tropical Southern Hemisphere (SH, Cape Grim Air Archive, CGAA)**
**and extra-tropical Northern Hemisphere (NH)**

To reconstruct the atmospheric history of $c$-$C_4F_8$ in the extra-tropical SH, 41 unique CGAA samples (collected 1978–2009, Langenfelds et al., 2014) were measured at CSIRO in 2011 (Ivy et al., 2012). Three CGAA tanks were measured at the beginning, in the middle, and towards the end of the measurements at CSIRO, with agreements within typical precisions or better (0.01–0.02 ppt). In addition, 8 SH samples were measured at SIO which were sub-sampled from CGAA parent tanks (fill dates 1986–2008, 0.60–1.17 ppt) into evacuated stainless steel (SS) tanks (4.5 L, Essex Industries, USA) with a vacuum manifold and pressure regulator shown not to produce any $c$-$C_4F_8$ artefacts. They were measured at SIO on Medusa 7 to take advantage of the more sensitive MSD and to evaluate the agreement with Medusa 9 measurements at CSIRO. Four of these CGAA subsamples measured at SIO agreed within precisions (delta mole fractions, $\Delta x = 0.00$–$0.01$ ppt, ratio = 1.0047, $R^2 = 0.9994$) with their CGAA parents measured at CSIRO, 2 subsamples showed a larger differences (0.018 and 0.027 ppt). The measurements of the seventh subsample and its CGAA parent were rejected, perhaps due to problems during the subsampling or with the parent tank. While we did not measure the CGAA parent of the eights subsample at CSIRO, we found agreement ($\Delta x = 0.01$ ppt) with another CGAA tank of similar air age ($\Delta t = 63$ days) measured at CSIRO. Four additional SH samples (fill dates 1995–2010, 0.84–1.25 ppt) were measured at SIO. Three were also in very good agreement with CGAA samples of similar fill date measured at CSIRO ($\Delta x < 0.006$ ppt, $\Delta t = 7$–$23$ days) and one showed a larger difference ($\Delta x = 0.05$ ppt). Based on an iterative filtering process designed to reject outliers greater than $2\sigma$ deviations from curve fits through the results for all 60 SH samples (41 at CSIRO and 19 at SIO) and pollution filtered monthly mean measurements (O'Doherty et al., 2001; Cunnold et al., 2002) at the extra-tropical stations CGO and ASA (Australia), 13 SH samples were rejected as outliers, leaving 47 SH samples (78 %).

To reconstruct the atmospheric history in the extra-tropical NH, 126 unique air samples mostly filled at SIO and THD (1973–2016) were measured at SIO. Additionally, 3 NH samples (filled in 1980 and 1999) were measured at CSIRO. Two of these tanks measured at CSIRO were filled together at SIO in 1999 with 2 tanks measured at SIO and the agreement is excellent ($\Delta x = <0.007$ ppt). The third tank, filled in 1980 at Cape Meares, Oregon, agreed within 0.034 ppt with another NH tank (filled at SIO within 9 days) measured at SIO. Despite this larger difference, the overall good agreement of NH and SH samples measured at SIO and CSIRO shows that measurements on the involved instruments were comparable and that calibration scales were properly propagated. Most of the NH samples had been filled during baseline conditions for various purposes using modified diving compressors (RIX, SA-3 and SA-6, Weiss and Keeling laboratories) and did not show any artefacts for many gases (e.g., Mühle et al., 2010; O'Doherty et al., 2014; Vollmer et al., 2016). For $c$-$C_4F_8$, however, comparisons with concurrent in situ measurements at MHD, THD, SIO and JFJ revealed artefacts for most of these samples and the iterative filtering process only retained $c$-$C_4F_8$ data for eleven NH samples. In contrast, CGAA tanks had been filled with a cryogenic method which did not produce any bias. Due to the sparse NH data and poor data quality before in situ measurements started in the NH, the fits used for the iterative filtering process of NH data had to be guided by the final SH fit shifted by 1.5 years to allow for the delay of $c$-$C_4F_8$ accumulation between the SH and NH due to inter-hemispheric transport (Mühle et al., 2010; Vollmer et al., 2016). Without this guidance, initial NH fits were dominated by high outliers, resulting in bad fits. Fig. 1 shows the filtered data and the final suggested fits.





### 2.3 Air extracted from firn

To augment the data set of in situ and archived air measurements, we measured $c$-$C_4F_8$ in samples from a subset of the firn sites described in Trudinger et al. (2016), namely NEEM08 in the NH and DSSW20K and SPO01 in the SH, plus one new site in the NH, Summit13, Greenland. We used the CSIRO firn model (Trudinger et al., 1997; Trudinger et al., 2013) to characterize the age of the air in these samples (detailed in Sect. 4.1). Here, we give a brief description of the firn sites. For a full description of the calibration of the CSIRO firn model for NEEM08, DSSW20K, and SPO01 see Trudinger et al. (2013), for Summit13 see Fig. S1.

NEEM08: Firn air was extracted from the EU borehole in July 2008 in northern Greenland, drilled near the North Greenland Eemian Ice Drilling Project (NEEM) deep ice core drilling site (77.45° N, 51.06° W) (Buizert et al., 2012). This site has a moderate snow accumulation rate of 199 kg m$^{-2}$ yr$^{-1}$.

Summit13: Firn air was collected in May 2013 at Summit, Greenland from a borehole (72.66° N, 38.58° W) drilled 10 km NNW of Summit Station, Greenland. The US Firn Air system (Battle et al., 1996) was used to extract the air from 19 depth levels in the firn from surface to just above bubble close-off (80.06 m). The 3 in borehole was drilled with the Eclipse Ice Drill (IDDO) and new rubber bladders (1/8 in thick) were fabricated (Greene Rubber Co., Woburn, MA) for use in this campaign. 2.5 L glass flasks were filled at all depths for high resolution measurements of gases performed by the National Oceanic and Atmospheric Administration (NOAA) ($CO_2$, $CH_4$, CO, $N_2O$, $SF_6$, $H_2$). Larger volume samples from pre-selected depth levels were filled in 35 L electropolished SS tanks using a KNF Neuberger pump (neoprene diaphragms). These samples were measured at SIO for $c$-$C_4F_8$ and a other trace gases (including $CH_4$, $N_2O$, CFCs, HFCs, HCFCs, and $SF_6$). For quality control purposes, the sample line was measured on site for $[CO_2]$ and $[CH_4]$ by CRDS (Los Gatos Research, μ-GGA) and [CO] by a Reducing Gas Detector (Peak Labs, RCP1) prior to filling the flasks. Summit has a moderate snow accumulation rate of 211 kg m$^{-2}$ yr$^{-1}$. CSIRO firn model calculations for Summit use the density profile from Adolph and Albert (2014) and mean annual temperature and pressure of 241.75 K and 665 mbar. The diffusivity profile and related parameters were calibrated using the measurements described above of $CO_2$, $CH_4$, $N_2O$, $SF_6$, CFC-11, CFC-12, CFC-113, $CH_3CCl_3$, HFC-134a, HCFC-141b, and HCFC-142b. Firn model results for these tracers are shown in Fig. S1.

DSSW20K: Firn air was collected in January 1998 in Eastern Antarctica (66.73° S, 112.83° E) from a borehole drilled 20 km west of the deep Dome Summit South (DSS) drill site near the summit of Law Dome (Smith et al., 2000; Sturrock et al., 2002; Trudinger et al., 2002). This site has a short firn column and a moderate snow accumulation rate of 150 kg m$^{-2}$ yr$^{-1}$.

SPO01: We only measured one sample collected in 2001 from 120 m from a borehole at the South Pole, Antarctica (90° S, 119° W) (Aydin et al., 2004; Sowers et al., 2005). This site has a deep firn column and a low snow accumulation rate of 75 kg m$^{-2}$ yr$^{-1}$, resulting in old firn air.

Firn air extracted from the DSSW20K, NEEM08, and SPO01 sites was measured at CSIRO in 2012 (Medusa 9), while Summit13 firn air was measured at SIO (Medusa 7), see Table 1. $c$-$C_4F_8$ firn measurement data are included in the data file listed in the Supplement. Other gases such as $CH_4$ and $N_2O$ were measured as well.



### 2.4 Air samples collected over India and the Indian Ocean


Air samples were collected on-board the UK FAAM (Facility for Airborne Atmospheric Measurements) BAe-146
aircraft during eleven flights conducted from June 12, 2016 to July 9, 2016 (9–28° N, 72–86° E) into 3 L pre-
evacuated electropolished SS flasks (SilcoCan, Restek) sealed with metal bellow valves (SS-BNVVCR-4,
Swagelok). During the time it took to compress the air samples to 41 psig (30–60 s depending on altitude) using a
metal bellows pump (PWSC 28823-7, Senior Aerospace, USA), the aircraft travelled ~7 km. Nine flights occurred
over Northern India and two over Southern India and the Indian Ocean. In total, 176 flask samples were collected,
with the majority (>90 %) of these samples filled below 1.5 km altitude. The size of the subsamples analyzed with
the Medusa 21 at University of Bristol was reduced to 1.75 L (from 2 L) and the sampling rate to 50 ml min$^{-1}$ (from
100 ml min$^{-1}$) to allow for triplicate analyses of each flask and to accommodate for the lower flask pressure. $c$-C$_4$F$_8$
measurements are reported on the SIO-14 calibration scale. Detection limits, blanks, and precisions were similar to
those stated above. For further details, see Say et al. (2019).

### 3 Bottom-up emission inventories (UNFCCC, EDGAR, NIRs, WSC)


Emissions of compounds, such as $c$-C$_4$F$_8$, into the atmosphere are often estimated by so called "bottom-up" methods,
which are based on information such as purchased, produced or imported amounts, industrial activities referred to as
activity data and estimated emissions factors for each emissive process. Developed countries report annual emissions
of GHG, including $c$-C$_4$F$_8$, and ozone depleting substances (ODS) to the UNFCCC using such bottom-up methods.
These data are however, by definition, not representative of total global emissions as developing countries do not
have the same comprehensive UNFCCC reporting requirements, including countries such as South Korea, China,
and Taiwan with sizable electronics and PTFE manufacturing capacities and thus with potentially significant $c$-C$_4$F$_8$
emissions. An additional complication is that several countries report unspecified mixes of PFCs or of PFCs and
HFCs and other fluorinated compounds, making it difficult or impossible to estimate emissions of individual
compounds, such as $c$-C$_4$F$_8$. In the Supplement, we gather available inventory information from submissions to
UNFCCC, National Inventory Reports (NIRs), the Emissions Database for Global Atmospheric Research (EDGAR),
the World Semiconductor Council (WSC), and the US Environmental Protection Agency (EPA) in an effort to
estimate contributions from unspecified mixes and countries not reporting to UNFCCC to compile a meaningful
bottom-up inventory. Globally these add up to 10–30 t yr$^{-1}$ (0.01–0.03 Gg yr$^{-1}$, 1 t = 0.001 Gg) from 1990 to 1999,
30–40 t yr$^{-1}$ (0.03–0.04 Gg yr$^{-1}$) from 2000 to 2010, and 100–116 t yr$^{-1}$ (~0.1 Gg yr$^{-1}$) from 2011 to 2014 (with a
substantial fraction due to the U.S. emissions from fluorocarbon production reported by US EPA). Similar to what
had been pointed out by Saito et al. (2010) and Oram et al. (2012), we will show in Sect. 5.2 and 5.3 that
measurement based ("top-down") global and most regional emissions are significantly larger than the compiled
bottom-up inventory information (see Fig. 5), similar to other PFCs (e.g., Mühle et al., 2010), reflecting the
shortcomings of current emission reporting requirements and emission inventories.



## 4 Modelling studies

### 4.1 CSIRO firn model

The CSIRO firn model and its use in global inversion frameworks has been described in detail (Trudinger et al., 2013; Trudinger et al., 2016; Vollmer et al., 2016; Vollmer et al., 2018a; Vollmer et al., 2018b). Air samples taken far away from pollution sources represent the background atmospheric trace gas composition at that time. Once air enters the firn vertical diffusion and other physical processes in the firn lead to mixing of air of different ages. Therefore, air extracted from firn must be described with an age distribution. We used the CSIRO firn model to describe the relationship between trace gas mole fractions measured in each extracted air sample from a given depth and the corresponding age distribution of high-latitude atmospheric mole fractions. The diffusion coefficient of $c$-$C_4F_8$ relative to that of $CO_2$ in air at 253 K used here was 0.47 with an estimated uncertainty of ~10 %. This value was determined using Equation 4 from Fuller et al. (1966) with Le Bas volume increments (e.g. Table 1.3.1, Mackay et al. (2006) and a multiplier for the Le Bas increments of 0.97 (which minimizes the difference of calculated relative diffusion coefficients of a number of compounds from values measured by Matsunaga et al. (1993, 2002, 2005)).

Figure 2 shows the measured depth profile of $c$-$C_4F_8$ (ppt) in air extracted from polar firn sites in the NH (Greenland) and the SH (Antarctica), for site details see Table 1. All samples showed $c$-$C_4F_8$ mole fractions above the detection limit. The firn reconstructed depth profiles are discussed in Sect. 4.3.1.

### 4.2 AGAGE 12-box model of the global atmosphere

The AGAGE 12-box two-dimensional model (Cunnold et al., 1983; Cunnold et al., 1997; Rigby et al., 2013) describes the transport and loss of trace gases in the global atmosphere. The model divides the atmosphere into four latitudinal bands at 0° and 30° S/° N and three altitude bands at 500 hPa and 200 hPa and calculates the mole fractions in each box. The AGAGE background sites (MHD, THD, RPB, SMO and CGO, see Table 1) were historically chosen to represent the trace gas mole fractions in the four lower (tropospheric) model "boxes". Model transport parameters were varied seasonally, but repeated annually. Given the very long atmospheric lifetime of $c$-$C_4F_8$ compared to the study period, the lifetime of $c$-$C_4F_8$ was assumed to be infinite in the model.

### 4.3 Global inversion methods

We used the AGAGE 12-box model in two different Bayesian inversions, denoted as the "CSIRO" and "Bristol" inversions, to estimate historic $c$-$C_4F_8$ emissions from our observations and to reconstruct historic abundances. Both inversions used in situ and archive data and the CSIRO inversion additionally used firn data. The observations need to be representative of clean background air at each sampling location. For in situ data, the AGAGE statistical method was used to remove pollution events and to calculate pollution-free monthly mean background air mole fractions for each AGAGE station (O'Doherty et al., 2001; Cunnold et al., 2002). As explained in Sect. 2.2, an iterative filtering algorithm starting out with all the archived air data and the pollution-free monthly means was then used to reject outliers for the extra-tropical SH and NH, mostly from the NH archive data. Due to the remoteness of the firn sample sites, we assumed background conditions without any filtering.



### 4.3.1 CSIRO inversion

The CSIRO inversion was developed to infer annual emissions at the global scale from firn, ice core and atmospheric measurements (Sturrock et al., 2002; Trudinger et al., 2002; Trudinger et al., 2016). Green's functions from the CSIRO firn model were used to relate the measured air in the firn samples to air in the atmosphere in the past, and Green's functions from the AGAGE 12-box model were used to relate global emissions with a specified latitudinal distribution to mole fraction in the extra-tropical SH and NH. The inversion included constraints to avoid negative mole fractions, negative emissions and unrealistic changes in emissions; these constraints were required due to the characteristics of inverting firn data and sparse archive data. The uncertainty in reconstructed mole fractions and inferred emissions was calculated using a bootstrap method that included the uncertainty in firn measurements, annual mean mole fraction (this uncertainty is temporally-correlated, see Supplement in Vollmer et al., 2018a), calibration scale (±2 %), and the firn model through the use of an ensemble of Green's functions corresponding to different firn model parameters (Trudinger et al., 2013; Trudinger et al., 2016; Vollmer et al., 2016).

Figure 3 shows the data that were used in the CSIRO inversion: annual values based on 10-year smoothing spline fits (i.e. 50 % attenuation at periods of 10 years) to monthly means of pollution free in situ measurements at the AGAGE background sites CGO (SH) and MHD (NH), annual values based on 10-year smoothing spline fits to measurements of the CGAA and archived NH air samples, and air extracted from polar firn in both hemispheres. Annual means from the spline were only used in the inversion when there were pollution free archive or in situ measurements around that time. Figure 3 also shows the final reconstructed abundances for the extra-tropical SH (solid black line) and NH (dashed black line) based on the optimized emissions. The measured mole fractions in firn air are plotted against their effective atmospheric ages if that age is after 1965, where the effective ages are calculated using the reconstructed history of atmospheric mole fractions determined by the CSIRO inversion (Trudinger et al., 2002). Before 1965, the growth rate in the atmosphere was small and uncertain; this makes it difficult to determine effective ages, so the earlier firn measurements are plotted against their mean ages. Firn depth profiles for each firn site corresponding to the CSIRO inversion results are shown in Fig. 2 (solid lines) and they typically agree with the measurements within precisions (1σ, shown as error bars).

Overall, the abundances reconstructed with the CSIRO inversion agree very well with the measurement data (also see Fig. S2). In Fig. S3, we show the effect of excluding different sites from the inversion and the sensitivity of the inversion to the relative diffusion coefficient of $c$-$C_4F_8$.

It should be pointed out that the deepest NEEM08 firn air sample for the NH showed slightly lower mole fractions (0.0085 ppt) than the deepest DSSW20K samples for the SH (0.021 ppt and 0.0185 ppt), although the mean ages are similar (1930s). The same applies to the second deepest NEEM08 (0.0105 ppt) and DSSW20K (0.018 ppt) samples (1940s), which is unexpected for a long-lived anthropogenic compound predominantly emitted in the NH. While the differences seem significant within the nominal precisions (0–0.0014 ppt) achieved for these firn samples measured only 1–2 times, they are not significant within typical precisions achieved for archive samples (~0.01–0.02 ppt) which are typically measured 3 or more times and these data are just at or below the typical detection limits of 0.01–0.03 ppt. Based on the order in which the firn samples were measured and the absence of detectable blanks, it seems unlikely that a small blank, memory, calibration, or measurement problem could have caused this small discrepancy. The early part of the reconstructed record, with near zero mole fractions, is also most susceptible to small



uncertainties in the calibrated diffusivity profiles versus depth for all sites used in the firn model, uncertainties in the
firn model structure (e.g., physical properties being invariant of time), or uncertainties in the diffusivity of different
tracers relative to each other. Thus, there are a number of possible reasons for the higher mixing ratio in the SH firn
data at this time, and we do not interpret this as evidence of higher mole fraction in the SH in the 1930s or 1950s.

### 4.3.2   Bristol inversion

The Bristol inversion was used to estimate annual fluxes of $c$-$C_4F_8$ using archive and in situ observations only (Rigby
et al., 2011; Rigby et al., 2014; Vollmer et al., 2018b). A priori, it was assumed that emissions were constant from
year to year, with an uncertainty in the year-to-year growth rate of 200 t yr$^{-1}$ (0.2 Gg yr$^{-1}$), approximately twice the
bottom-up estimate in Sect. 3. The derived emissions uncertainties include contributions from the measurement
repeatability, the calibration scale uncertainty, and the model-measurement representation error (Rigby et al., 2014).
Furthermore, because some archive air samples exhibit substantial short-timescale (< 1 year) variations that are
unlikely to represent real changes in the background atmosphere (Fig. 1), the minimum uncertainty was set to the
maximum deviation of the archive air samples from the smooth curve in Fig. 1 (0.03 ppt). Model representation
errors were estimated as the variability of the pollution-free monthly baseline means determined by the AGAGE
pollution algorithm (O'Doherty et al., 2001; Cunnold et al., 2002) from the high-frequency in situ data at each station
for each given month. For periods without in situ data, the representation error was assumed to be equal to the
average baseline variability from in situ data in the same latitudinal band scaled by the measured $c$-$C_4F_8$ abundance.
The calibration scale propagation uncertainty is estimated based on propagation uncertainties of the $c$-$C_4F_8$
calibration scale from primary gravimetric standards to secondary standards within the "R1" relative calibration
framework used in AGAGE and on propagation uncertainties from the R1 framework to the standards used to
measure individual samples. Figure 4 shows that there is good agreement between the archived air samples (Sect.
2.2) and the pollution free monthly mean in situ data from the AGAGE background sites (MHD and THD, RPB,
SMO, and CGO) used in the Bristol inversion and the reconstructed mole fractions for the four latitudinal bands
which these samples represent (see also Fig. S4).

### 4.4 Regional model and inversion study using NAME-HB for eastern Asia

To investigate regional emissions in eastern Asia (20° N–50° N and 110° E–160° E) from our observations we used
an inversion method based on Bayesian inference. We estimated annual mean emissions, assuming that emissions are
constant in both space and magnitude during each calendar year. Here, the inversion used observations from the
Gosan station as this site was operated with relatively few interruptions from October 2010 to the end of 2017, with
best data coverage from 2011 to 2015. These observations were binned into 12 hourly averages. The inversion
method requires an atmospheric transport model to derive the sensitivity of the observations to a surface emissions
field. We used the Lagrangian NAME (Numerical Atmospheric dispersion Modelling Environment) model from the
UK Met Office (Jones et al., 2007), driven by meteorology from the Met Office Unified Model (Walters et al., 2014).
The sensitivity was derived by releasing 20,000 hypothetical air parcels per hour of measurement from Gosan
station, which were transported backwards in time for up to 30 days. The model recorded the time and location that
air parcels interacted with the surface (below 40 m above ground level at a spatial resolution of 0.352° by 0.234°),





and these data were used to form an aggregated 30-day sensitivity or "footprint" map for each hour of measurement.
In addition, the model recorded the time and location that air parcels left the domain boundaries to provide the
sensitivity to the boundary conditions. The footprint maps, generated over the domain 5° S–74° N and 55° E–192° E
and up to 19 kilometres, were aggregated into 12 hourly averages.
We used a trans-dimensional hierarchical Bayesian method (NAME-HB) with a Metropolis-Hastings Markov chain
Monte Carlo (MCMC) algorithm (Metropolis et al., 1953; Hastings, 1970) to solve the inverse problem. This
allowed spatial emission estimates of $c$-$C_4F_8$ to be derived, whilst considering the uncertainties in the model,
measurements and a priori information and importantly the uncertainty in these uncertainties. Bayesian methods
require a priori knowledge, here the emissions and boundary conditions. As little information on eastern Asia's $c$-
$C_4F_8$ emissions (see Sect. 3) was available, we based our mean a priori emissions on those estimated by Saito et al.
(2010). We spread their emissions for each reported country uniformly over the area of each country, rather than use
population density (as in Saito et al., 2010) as that is not likely to be a good proxy of $c$-$C_4F_8$ emissions. We also
spread 0.11 Gg yr$^{-1}$ of emissions over the rest of the domain where the footprint was calculated. The value of 0.11 Gg
yr$^{-1}$ is an approximate scaling of the global total emissions based on population in this outer domain, i.e. the
remainder of the domain not defined as eastern Asia. We do not report emission estimates outside of eastern Asia
due to large posterior uncertainties, but including them assisted with determination of the boundary conditions (or
non-proximal emissions). We assigned a large uncertainty to these a priori emissions (Table S1), which were
governed by a log-normal distribution, so that they were uninformative and the observations dominated the
estimation. We set a priori boundary conditions to be the mean background mole fractions measured at MHD on
each vertical boundary (N, E, W, S) of the NAME domain. Offsets to the boundary conditions on each boundary
were estimated in the inversion on a monthly basis.
The hierarchical nature of the inversion method means that hyper-parameters were also incorporated to include
uncertainties in the NAME sensitivities, which are described by a multivariate Normal distribution (see Ganesan et
al., 2014). The reversible jump, or trans-dimensional, aspect of the inversion means that the underlying resolution at
which the emissions are estimated is itself explored during inference (Lunt et al., 2016). Table S1 shows the a priori
probability distributions assigned to the emissions and boundary conditions scaling factors, model uncertainty and
underlying grid. The posterior emissions estimates and their uncertainties were governed by exploring the spaces of
each of these parameters and hyper-parameters.
**4.5 Regional model and inversion study using InTEM for Western Europe**
To investigate regional emissions in Western Europe (36° N–66° N and -14° E–31° E) we used InTEM, an inversion
framework (Arnold et al., 2018) based on the NAME Lagrangian transport model (Jones et al., 2007), together with
observations from MHD, Tacolneston (TAC), Jungfraujoch (JFJ) and Monte Cimone (CMN). A priori estimates
were considered unknown (see Sect. 3 and the Supplement) and therefore set to a uniform distribution of 0.2 Gg yr$^{-1}$
over the whole land area within the inversion domain with an uncertainty of 0–0.62 Gg yr$^{-1}$. Observational
uncertainty was time varying and estimated as the variability of the observations in a 6 hour moving window plus the
measurement repeatability determined from repeat measurements of the on-site calibration standards. Model
uncertainty was estimated every 2 hours as the larger of the median of all pollution events at each station in a year or



16.5 % of the magnitude of the pollution event. A temporal correlation of 12 hours was assumed in the model
uncertainty at each station. An analytical solution was found that minimized the residual between the model and the
observations and the difference between the posterior and the a priori flux estimate, balanced by the uncertainties of
both. The baseline was estimated in the inversion following Arnold et al. (2018). The variable resolution of the
inversion grid was calculated and refined within InTEM based on the magnitude of the footprint and emissions from
each grid box. The inversions were run 24 times per year, each time with a randomly generated sub-sample (90 %) of
the available observations from each station (10 % removed in 5-day blocks), to further explore the uncertainty.
Emissions and uncertainties were averaged across the 24 individual inversions thereby assuming 100 % correlation
between uncertainties in these separate inversions. 1-year inversions were performed covering the period 2013–2017.
**4.6 Regional model and inversion study using NAME-HB for India**
To investigate regional emissions from the Indian subcontinent from the samples taken on-board a research aircraft
in June and July 2016 (see Sect. 2.4) we used the NAME-HB inversion method described in Sect. 4.4 and Table S1.
Here, the domain spanned from 6° N to 48° N and from 55° E to 109° E with an altitude up to 19 kilometers and
emissions were estimated as the mean over the 2-month period. As with eastern Asia and Western Europe studies,
the sensitivity of the atmospheric measurements to surface emissions was derived using the NAME model. Back-
trajectories were simulated for each minute of each flight path for up to 30 days backward in time. To account for the
motion of the aircraft, hypothetical air parcels were released from a cuboid whose dimensions were defined as the
change in latitude, longitude and altitude of the aircraft during each 1 minute period, at a release rate of 1000 air
parcels min$^{-1}$. Wherever possible, samples were collected during periods of level flight, to minimise the altitude
component of the release volume. India's a priori emissions were set to 18 % of global $c$-$C_4F_8$ emissions (from Sect.
5.2), equal to India's fraction of the global population, but uniformly distributed over India. A large uncertainty was
assigned (Table S1) to reflect the lack of information on India's current $c$-$C_4F_8$ emissions. A priori vertical boundary
conditions were assigned using background mole fractions from MHD (N, E and W) and CGO (S). Offsets to these
boundary conditions were estimated in the inversion. We only report emissions for Northern and Central India (NCI)
as the inversion has low sensitivity over southern India and Sri Lanka and the north western edge of the domain and
no sensitivity beyond the Himalayas (see Fig. S5). Sensitivity tests indicate that $c$-$C_4F_8$ emissions determined for
NCI are insensitive to the choice of a priori emissions (see Fig. S6).
**4.7 Pollution events at Zeppelin station**
The Zeppelin (ZEP) station is located in a clean Arctic environment and receives air masses representative mostly of
the Arctic background. Nevertheless, 10 cases of enhanced $c$-$C_4F_8$ mole fractions were observed with the arrival of
air masses from Eurasia. To trace the origin of these events, we used 3-hourly 50-day backward simulations for a
passive tracer with version 10 of the Lagrangian particle dispersion model FLEXPART (Stohl et al., 2005). The
model was driven with operational meteorological analyses of the European Centre for Medium Range Weather
Forecasts (ECMWF, https://www.ecmwf.int/). The model set-up was similar to that typically used for inversion
studies (Stohl et al., 2009), but the number of events observed at the station was too small for a sensible regional
inversion. Instead, we inserted unit emission sources (~1 kg s$^{-1}$) at two facilities in Russia producing PTFE and





halogenated chemicals including $c$-$C_4F_8$ (HaloPolymer, Kirovo-Chepetsk, Kirov Oblast and Galogen Open Joint-
Stock Company, Perm), one or both of which we suspect to be responsible for most of the observed enhancements.
We then scaled the modeled $c$-$C_4F_8$ mole fractions based on these two unit sources to the observed enhancements to
estimate the source strength required to explain the observations. The two sources are quite close to each other and
thus very much correlated so it was impossible to quantify the influence of each source individually, but it turned out
that each source required about the same flux to produce a similar good match with the observations.

## 5 Results and discussion

### 5.1 Atmospheric histories of $c$-$C_4F_8$ in both hemispheres

Figure 1 shows the atmospheric histories of $c$-$C_4F_8$ in the extra-tropical NH and SH determined from several sets of
archive measurements and pollution filtered data from six in situ measurement stations. As detailed in Sect. 2.2, the
data shown have gone through an iterative filtering process which mostly removed outliers from the NH record. The
pollution free monthly mean in situ data for the four extra-tropical NH stations shown here and ZEP agree within
precisions, although JFJ data tends to be at the lower range since early 2015 for unknown reasons. The two extra-
tropical SH stations, CGO and ASA also agree well with each other. Mole fractions measured in both hemispheres
show a clear and consistent interhemispheric gradient reflecting the high precision of the measurements and
indicating that emissions of $c$-$C_4F_8$ predominantly occur in the NH. These data form a consistent atmospheric record
of $c$-$C_4F_8$ from the late 1970s to 2017 in both hemispheres, albeit with very sparse data for the NH before in situ
measurements started at JFJ and at other NH stations.
To augment our $c$-$C_4F_8$ data set and to extend our reconstruction further backwards in time, we measured air samples
extracted at several firn sites from both hemispheres and interpreted the data with the CSIRO global inversion
framework. The CSIRO inversion (see Sect. 4.3.1) yields the atmospheric history of $c$-$C_4F_8$ starting in 1900 until
present, although abundances are essentially not different from zero (<0.02 ppt) until the early 1960s (Fig. 3).
Average global $c$-$C_4F_8$ mole fractions reached 0.45 ppt in 1980, 0.74 ppt in 1990, 0.97 ppt in 2000, 1.29 ppt in 2010,
and 1.66 ppt in 2017. The Bristol inversion (see Sect. 4.3.2) does not incorporate firn data, still atmospheric histories
of the two inversions generally are in good agreement (see Fig. S7).
The CSIRO inversion reconstructs that the global rise rate of $c$-$C_4F_8$ accelerated from near zero before the late 1960s
to ~0.03–0.04 ppt yr$^{-1}$ in the mid-1970s to late 1980s, after which the rise slowed down to ~0.02 ppt yr$^{-1}$ in the early
1990s to mid-2000s. It increased again in the early 2000s and reached ~0.07 ppt yr$^{-1}$ in 2017.
Compared to Oram et al. (2012), our work extends the SH record from 2008 until present and, arguably, from 1978
back to 1900. Furthermore, it adds the full NH record. SH mole fractions reconstructed by Oram et al. (2012) are
very similar in 1978 and 1990, but ~0.06 ppt lower in the mid-1980s (~11 %) and the late 1990s to late 2000s (~5 %,
see Fig. S7). Although the stated precision in Oram et al. (2012) of 0.8 % (~0.01 ppt at 1.2 ppt) is similar to the 0.01–
0.02 ppt achieved here, the resulting precisions of the CGAA measurements achieved here are significantly
improved, e.g., the noise in the CGAA reconstruction by Oram et al. (2012) is about as large as the interhemispheric



gradient determined here (see Fig. S7). The estimated accuracy of the SIO-14 $c$-$C_4F_8$ calibration scale of ~2 % also
compares favorably to previous calibration scale uncertainties.

## 5.2 Global $c$-$C_4F_8$ emissions

Global $c$-$C_4F_8$ emissions (Fig. 5 and Supplement) started to increase in the early 1960s (CSIRO inversion) from near
zero to ~1.2 Gg yr$^{-1}$ in the late 1970s to the late 1980s. The Bristol inversion initially reconstructs lower emissions,
but catches up by the early 1980s, perhaps because firn data were not incorporated. After this, emissions determined
by both inversions declined to ~0.8 Gg yr$^{-1}$ in the mid-1990s to early 2000s. Since then emissions kept increasing,
reaching ~2.2 Gg yr$^{-1}$ in 2017. Both inversions reconstruct emissions which are significantly larger than available
bottom-up inventory information (see Sect. 3 and the Supplement), reflecting the shortcomings of the current
UNFCCC reporting requirements and inventories.
Emissions presented by Oram et al. (2012) agree very well from 2001 to 2007 with our results and on average also
from 1978 to 2001, although they show larger variability. Global emissions roughly estimated by Harnisch (2000)
based on measurements by Travnicek (1998) of ~0.7 Gg yr$^{-1}$ from 1978 to 1997 are lower than our estimate of $1.01 \pm$
$0.10$ Gg yr$^{-1}$. Saito et al. (2010) estimated global emissions of $0.6 \pm 0.2$ Gg yr$^{-1}$ from January 2006 to September
2009, about half of our $1.16 \pm 0.09$ Gg yr$^{-1}$ estimate. This is likely due to slowly changing $c$-$C_4F_8$ mole fractions in
calibration tanks used by NIES (Takuya Saito, personal communication, 2018), which would significantly affect the
background rise rate and thus global emissions, but would have had less influence on the regional emissions
estimated by Saito et al. (2010) as these are mostly dependent on the magnitude of the much larger pollution events
above background.
Global emissions of $c$-$C_4F_8$ have clearly not levelled off at 2005–2008 levels as had been suggested by Oram et al.
(2012), but kept rising. In contrast, emissions of other minor PFCs, $C_2F_6$ and $C_3F_8$, have decreased since the early
2000s and stabilized in recent years (Trudinger et al., 2016), reflecting that emission sources and/or use patterns of $c$-
$C_4F_8$ are different from those of the other minor PFCs. Weighted by GWP estimated 2017 emissions of $c$-$C_4F_8$, $C_3F_8$,
$C_2F_6$, and $CF_4$ were 0.021, 0.005, 0.022, and 0.083 billion tonnes of $CO_2$-eq., respectively (see Fig. S8). $c$-$C_4F_8$
emissions have been larger than those of $C_3F_8$ since 2004 and, assuming continued growth, will also outpace $C_2F_6$
emissions within a year or two, so that $c$-$C_4F_8$ will become the second most important PFC emitted into the global
atmosphere. In the next section, we will investigate regional emissions of $c$-$C_4F_8$ to gain a better understanding how
individual regions and sources may contribute to the global emissions.

## 5.3 Regional $c$-$C_4F_8$ emission studies

### 5.3.1    Emissions from eastern Asia

Within the AGAGE network, the two stations in eastern Asia, Gosan (GSN) and Shangdianzi (SDZ), show by far the
most frequent and most pronounced pollution events of up to ~14 ppt above NH background, indicating significant
regional emissions (see Fig. S9). Therefore, we use a regional inverse method (NAME-HB) to infer the emissions in
this region (20° N–50° N and 110° E–160° E, see Sect. 4.4). We focus on the observations from GSN as this site was
operated with relatively few interruptions from June 2010 to the end of 2017 and had almost full coverage for each



year from 2011 to 2015. Significantly longer data gaps exist for SDZ, which would have made interpretation of
inversion results more difficult. The sensitivity of the inversion generally decreases with distance to the receptor
station resulting in relatively low sensitivity for emissions from western China, eastern Japan and Taiwan (the
cumulative footprint map for 2010–2017 is shown in Fig. S10). Therefore, we report in Table 2 and Fig. 6 estimated
emissions for eastern China, western Japan, South Korea, North Korea, and Taiwan. $c$-$C_4F_8$ emissions in this eastern
Asian domain increased from $0.36 \pm 0.07$ Gg yr$^{-1}$ in 2010 to $0.73 \pm 0.13$ Gg yr$^{-1}$ in 2016 and 2017 and were
dominated by emissions from eastern China. Compared to the a priori emissions for eastern China of 0.185 Gg yr$^{-1}$,
which are based on the Saito et al. (2010) estimate for all of China for November 2007 to September 2009, this
represents an increase of ~62 % in 2010 and more than a tripling in 2017. Note, that if we were to sum up emissions
for all regions of China, including those where the inversion has low sensitivity, total emissions would be another
~50–75 % higher. In contrast, the EDGAR 4.2 emission inventory, the only available bottom-up information (see
Sect. 3 and the Supplement), suggests no significant emissions from China.
For western Japan we find emissions of ~0.02 Gg yr$^{-1}$ (no trend), ~30 % lower than the a priori emissions (from Saito
et al. 2010, see Sect. 4.4). While total country emissions are likely higher, the available bottom-up information (see
Sect. 3 and Supplement) suggests 1 order of magnitude lower emissions for all of Japan. For South Korea, the
inversion adjusts emissions down to 0.01–0.02 Gg yr$^{-1}$ in most years and up to ~0.04 Gg yr$^{-1}$ in 2014 and 2015.
Except perhaps for 2012 and 2017, emissions from South Korea are significantly higher than the 0.003–0.008 Gg yr$^{-1}$
suggested by the available bottom-up information. Emissions from Taiwan show no trend and are relatively small
with ~0.01 Gg yr$^{-1}$, which is ~50 % of ~0.02 Gg yr$^{-1}$ indicated by the Taiwanese NIR, though it should be noted that
the inversion has relatively low sensitivities for some parts of Taiwan (see Fig. S10). Overall, emissions from
western Japan, South Korea, and Taiwan are small, despite their large semiconductor industries (see also Fig. 7),
suggesting that this industry sector is not a major emitter of $c$-$C_4F_8$. Emissions from North Korea are also small.
Combined regional $c$-$C_4F_8$ emissions doubled from 2010 to 2016, driven by Chinese emissions. They represent $31 \pm$
$4$ % of global emissions (2010–2017), while eastern China's emissions represent $28 \pm 4$ %. The difference between
global and eastern Asian emissions remained relatively consistent, ranging from ~1.04 Gg yr$^{-1}$ in 2010 to 1.47 Gg yr$^{-1}$
$^{-1}$ in 2017 with an average of $1.20 \pm 0.14$ Gg yr$^{-1}$ from 2010 to 2017 and $1.15 \pm 0.03$ Gg yr$^{-1}$ from 2011 to 2015, the
years with the best data coverage at GSN and thus highest confidence in the results. This means that the increase in
global emissions is essentially explained by the increase in eastern Asian emissions, i.e. mostly from China, but also
that significant emissions of ~1.16 Gg yr$^{-1}$ exist outside of the investigated region (a fraction of which may stem
from industries located in parts of China and perhaps Japan where the inversion has low sensitivity).
Figure 7 shows that from 2010 to 2017 emissions in eastern China occur from the highly industrialized provinces
Shandong, Tianjin, and parts of Henan and Hebei (south/southwest of Beijing) as well as from Shanghai and
neighboring Jiangsu (to the north), Anhui (to the west) and Zhejiang (to the south) in the Yangtze River Delta region.
Also shown are locations of potential industrial $c$-$C_4F_8$ point sources. For South Korea, western Japan and Taiwan,
semiconductor fabrication plants do not seem to be dominant $c$-$C_4F_8$ emitters as they are not co-located with large $c$-
$C_4F_8$ emissions (though the inversion has low sensitivity for eastern Japan, where many more FABS and several
PTFE and HCFC-22 plants are located, hence emissions from this region cannot be analyzed).





In China, the picture is less clear than in South Korea, Japan and Taiwan, as semiconductor fabrication plants in the
Yangtze River Delta region are co-located with strong $c$-$C_4F_8$ emissions, while those near Beijing are not. Many of
the potential production facilities of TFE and HFP monomers and PTFE and FEP polymers are co-located with areas
where strong $c$-$C_4F_8$ emissions occur. This is consistent with information from the second largest producer of PTFE
in China that they do not recover $c$-$C_4F_8$ by-product, but do emit $c$-$C_4F_8$ to the atmosphere (Hu, J., personal
communication, 2018). Still, the two facilities north east of Beijing do not seem to emit $c$-$C_4F_8$, perhaps reflecting
that some producers minimize $c$-$C_4F_8$ emissions, e.g., to increase yield or to use $c$-$C_4F_8$ for other purposes, such as
for the semiconductor industry. Several facilities are also located in provinces for which the inversion has low
sensitivity. Most HCFC-22 production facilities are not co-located with strong $c$-$C_4F_8$ emissions, while $CHCl_3$
production facilities tend to be in areas with $c$-$C_4F_8$ emissions. This may reflect that $CHCl_3$ production has shifted
from use as feedstock to produce HCFC-22 for dispersive applications (refrigeration or foam blowing), where no $c$-
$C_4F_8$ emissions occur, to production of TFE/HFP/PTFE/FEP via HCFC-22 pyrolysis, where $c$-$C_4F_8$ by-product
emissions occur, perhaps at the same or close-by facilities. This would be consistent with the start of the HCFC
phase-out for dispersive applications in developing countries mandated by the Montreal Protocol on the Protection of
Ozone Layer. Then again, $CHCl_3$ has other uses, e.g. as solvent (Tsai, 2017), without any potential $c$-$C_4F_8$ emissions.
There is no strong correlation between $c$-$C_4F_8$ emissions distribution and population density, e.g. emissions from
Henan and Hebei provinces are significantly lower than those from Shandong despite similar total population, which
may indicate that combustion of fluoropolymers in waste incineration facilities (Morisaki, 1978; Kannan et al., 2005;
van der Walt et al., 2008; Ji et al., 2016; Bezuidenhoudt et al., 2017) is not a dominant source of $c$-$C_4F_8$ emissions.
If $c$-$C_4F_8$ emissions in eastern Asia are indeed predominantly associated with TFE/HFP/PTFE/FEP production via
the pyrolysis of HCFC-22, $c$-$C_4F_8$ emissions may co-occur with small emissions of HCFC-22, TFE and HFP. $CHCl_3$
and HFC-23 emissions may also co-occur as HCFC-22 is produced from $CHCl_3$ and HFC-23 is a by-product that in
developing countries is probably again vented to the atmosphere since the UNFCCC Clean Development Mechanism
(CDM) funding to avoid HFC-23 emissions has expired (Simmonds et al., 2018; Say et al., 2019). While the global
atmospheric lifetime of TFE is only ~2 days, the lifetime of HFP is ~6 days (Acerboni et al., 2001), so that HFP may
be detectable near strong emission sources and serve as a sensitive marker for regional TFE/HFP/PTFE/FEP
production. After adding HFP to the measurements in late 2018, we find strong HFP pollution events at SDZ which
are associated with $c$-$C_4F_8$, $CHCl_3$, HCFC-22 and HFC-23 pollution events. HFP pollution events at GSN are much
weaker, reflecting the short atmospheric lifetime and the more distant source region, but they are also associated with
$c$-$C_4F_8$, $CHCl_3$, HCFC-22 and HFC-23 pollution events. At both sites, however, $c$-$C_4F_8$ pollution events also co-
occur with enhancements of other anthropogenic compounds which may just point to generally polluted air in the
region, so it is difficult to draw definitive conclusions. Still it is clear that HFP is emitted in eastern Asia, likely in
China, and HFP as well as $c$-$C_4F_8$ are associated with PTFE/FEP production. Measurements of HFP at SIO and ASA,
confirm that it is virtually absent from the global background atmosphere even in urban environments.
Overall, the strong c-$C_4F_8$ emissions in eastern China and their source regions are consistent with emissions from
TFE/HFP/PTFE/FEP production facilities due to little or no recovery or abatement of $c$-$C_4F_8$ by-product and the
significant fraction of global PTFE production (53–67 % in 2015) in China (see Table S3).



### 5.3.2 Emissions from North Western Europe

Outside of eastern Asia, the TAC station in East Anglia, UK shows by far the most frequent and most pronounced $c$-$C_4F_8$ pollution events of any AGAGE station, with a few reaching ~5 to 10 ppt above NH background, indicating close-by emissions. Data from the TAC, MHD, JFJ and CMN stations and the InTEM regional inverse method (see Sect. 4.5) were used to estimate emissions from North Western Europe (42° N to 59° N and -11° E to 15° E) based on to the areas of highest sensitivity to the observations (see Fig. S11). Compared to eastern Asia, we find only small emissions of ~0.02 ± 0.01 Gg yr$^{-1}$ (2013–2017) without any significant temporal trend, corresponding to only ~1 % of global emissions, despite an estimated 14 % of global PTFE production in 2015 (see Table S3). The mean distribution of emissions is shown in Fig. 8. As in eastern Asia, most identified semiconductor FABS in Europe are not co-located with $c$-$C_4F_8$ emission hotspots, except perhaps several FABS in Northern France, the UK, and Ireland. Producers of PTFE and FEP and facility locations in Europe were determined from company websites (3M/Dyneon, AGC/Asahi Glass, Arkema, Chemours/DuPont, Saint-Gobin, Solvay) and the European Pollutant Release and Transfer Register (https://prtr.eea.europa.eu), but it is very difficult to determine at which of the many facilities PTFE or FEP are actually produced and thus where $c$-$C_4F_8$ may be emitted. It seems that several facilities in The Netherlands, Belgium, the UK, France, and Italy which likely produce PTFE are co-located with identified $c$-$C_4F_8$ emission hotspots (Fig. 8). Still, many mismatches exist, reflecting the uncertainties in determining the exact facility locations, the relatively small emission strength and uncertainties of the inversion. As in eastern Asia, there seems to be no correlation with population density, which suggests that waste incineration of fluoropolymers is not a dominant $c$-$C_4F_8$ source here either. While emissions are relatively small, it is noteworthy that UNFCCC reporting by The Netherlands, the UK, Belgium, and France suggest much smaller $c$-$C_4F_8$ emissions.

### 5.3.3 Emissions from South Eastern Australia

Other urban locations of the AGAGE network, such as SIO, USA and ASA, Australia show much smaller pollution events above global background (up to ~2.5 ppt) than those seen at TAC, suggesting even lower emissions. Still, the few pollution events at ASA and even CGO are interesting as production of PFCs in Australia has never been recorded. CFC-11, CFC-12, and HCFC-22 were manufactured starting in 1962 at two facilities in Sydney, but production ceased in 1995 and trace gas emissions from Sydney are rarely if ever observable at CGO or ASA. Without any currently known fluorocarbon production, any c-$C_4F_e$ pollution events observed at CGO or ASA should not be due to fugitive emissions. $c$-$C_4F_8$ imports to Australia are ~4 to 50 kg yr$^{-1}$ (2011–2015), likely for minor refrigeration uses. In contrast, small but identifiable $c$-$C_4F_8$ pollution episodes at CGO suggest Melbourne emissions of ~2 t yr$^{-1}$ (0.002 Gg yr$^{-1}$) in 2016 (down from ~5 t yr-1 in 2009, Inter Species Correlation method, ISC, c.f., Fraser et al., 2014; Dunse et al., 2018). Scaled by population to Australia (for lack of a better proxy), emissions from 2009 to 2016 could be ~10–25 t yr$^{-1}$ (0.01–0.025 Gg yr$^{-1}$), 2–3 orders of magnitude higher than import data suggests. Since early 2017, HFP has been measured at ASA. Occasional, small HFP pollution events, which are often, but not always, associated with $c$-$C_4F_8$ pollution events, may point to small scale production of PTFE/FEP/TFE/HFP in Melbourne or perhaps these small emissions stem from incineration of waste containing fluoropolymers. Another possible explanation could be that more $c$-$C_4F_8$ is imported in products for minor applications than identified in import data due to inadequate labelling. On a global scale, estimated Australian $c$-$C_4F_8$ emissions of ~0.015 Gg yr$^{-1}$





are small, ~0.7 % of global emissions. PFC ($CF_4$, $C_2F_6$) pollution episodes at Cape Grim and Aspendale due to PFC
emissions from South Eastern Australian aluminum smelters (Portland and Pt. Henry, Victoria and Bell Bay,
Tasmania) do not show any evidence of $c$-$C_4F_8$ emissions (Fraser et al., 2013; CSIRO unpublished data).
**5.3.4     Emissions from undersampled regions such as the US, India, Russia**
The AGAGE network does not closely monitor large areas of the globe where $c$-$C_4F_8$ emissions may occur. For
example, many semiconductor FABS are located in the western, southern, and eastern US and chemical facilities
located in the southern and eastern US are estimated to account for ~10 % of global PTFE production in 2015, while
facilities in India and Russia are estimated to account for ~8 % and ~6 %, respectively (see Tables S3 and S4). The
two AGAGE stations in California are only able to capture a fraction of these emissions due to predominant westerly
winds and therefore we cannot estimate $c$-$C_4F_8$ emissions from the continental US. If PTFE production facilities in
the US are operated as in NW Europe, emissions should be similarly small. If facilities in India and Russia are
operated as in China, emissions could be significant as well. In the case of Russia this seems likely as the original
technology for fluoropolymer production in China apparently stems from Russia (Buznik, 2009).
**5.3.5     Emissions from India**
Say et al. (2019) recently presented measurements from an aircraft campaign in June and July 2016 (see Sect. 2.4)
over the Indian subcontinent to determine emissions of ODS and HFCs. Here we use their $c$-$C_4F_8$ measurements and
the NAME-HB inversion (see Sect. 4.6) and estimate emissions of 0.14 (0.09–0.20) Gg yr$^{-1}$ for Northern and Central
India (NCI). Data are only available for two months in 2016, but seasonality in industrial emissions of $c$-$C_4F_8$ is not
expected. The posterior emissions distribution (Fig. 9) is consistent with emissions from facilities producing PTFE.
Several of the HCFC-22 production facilities are co-located or very close to these PTFE producing facilities,
suggesting that a fraction of HCFC-22 is pyrolyzed to produce monomers for PTFE and FEP. Two HCFC-22
production facilities are outside of areas with strong $c$-$C_4F_8$ emissions, possibly because these two sites focus on
production of HCFC-22 for dispersive applications (refrigeration or foam blowing), where no $c$-$C_4F_8$ emissions
occur. The single known FAB in India is not co-located with significant $c$-$C_4F_8$ emissions. As in eastern Asia and
North Western Europe, there is no apparent correlation of $c$-$C_4F_8$ emissions with population density. Emissions
predominantly occur outside of the Indo-Gangetic plain, the most densely populated region of India. The derived
emissions account for 6.8 (4.4–9.7) % of global $c$-$C_4F_8$ emissions in 2016, in comparison to the estimated ~8 % of
2015 global PTFE production capacity (see Table S3). Perhaps even clearer than in eastern Asia, these results point
to PTFE production as dominant emission source of $c$-$C_4F_8$. All known Indian PTFE manufacturers are located
within the NCI domain, hence the estimated emissions are likely to be roughly representative of India's national total,
though further atmospheric measurements would be required to confirm this.
**5.3.6     Emissions from facilities in Russia**
The ZEP site in remote Svalbard shows ten small $c$-$C_4F_8$ pollution events above NH background of up to ~0.4 ppt.
FLEXPART backward simulations could trace some of these events to two facilities in Russia which produce PTFE
and halogenated chemicals including $c$-$C_4F_8$ itself (HaloPolymer, Kirovo-Chepetsk, Kirov Oblast and Galogen Open




Joint-Stock Company, Perm). Figure S12 shows the FLEXPART footprint emission sensitivity map for the largest
observed $c$-$C_4F_8$ enhancement on November 19, 2016, suggesting direct transport from the two PTFE production
sites. The emission sensitivity maps indicate that for six of the ten observed pollution events the air had clearly
passed over one or both of these two sources, even though the timing of the observed events was often not well
matched by the model, which was sometimes off by up to about half a day. While this is not surprising given the
large distance between the source and the receptor, it means that the two sources could not be clearly separated,
especially since the FLEXPART emission sensitivity often also covered both sites for the same arrival times at ZEP.
Assuming a unit emission at those two locations and scaling the resulting simulated mole fractions at ZEP to the
observed enhancements above background we estimated the emission strength for the two sites together for each
event (see Sect. 4.7). Five of the ten pollution events could be approximately reproduced by this method and required
a flux of $0.18 \pm 0.06$ Gg yr$^{-1}$, while the sixth event required ~0.54 Gg yr$^{-1}$. Averaged for all six events $0.24 \pm 0.15$ Gg
yr$^{-1}$ would be required. Either of these fluxes would be significant, representing $9 \pm 3$ %, 26 %, and $12 \pm 7$ % of
global emissions, respectively, compared to ~6 % of estimated global PTFE production in Russia. The uncertainty of
this estimate is large because only a few events were observed and not all of them were reproduced equally well by
FLEXPART. Similar to eastern Asia, the largest $c$-$C_4F_8$ pollution event also showed enhancements of HCFC-22 and
HFC-23, pointing to the production of PTFE as source, but other halogenated compounds were also elevated.
**6 Summary and conclusions**
We determine the atmospheric histories of $c$-$C_4F_8$ (PFC-318, perfluorocyclobutane) in both hemispheres based on
measurements of archived, in situ, and firn air samples in conjunction with the CSIRO firn model, the AGAGE 12-
box model, and two global inversion frameworks. Compared to previous studies, our work extends the Southern
Hemisphere record from 1978 back to 1900 and from 2008 until 2017 and adds a Northern Hemisphere record, all
reported with better precisions for air archive measurements (~1–2 %) and a lower uncertainty (2 % versus ≤ 7 %) of
the SIO-14 gravimetric calibration scale. We find global $c$-$C_4F_8$ atmospheric mole fractions near zero (< 0.02 ppt)
from 1900 until the early 1960s, after which they rose sharply, reaching 0.45 ppt in 1980, 0.74 ppt in 1990, 0.97 ppt
in 2000, 1.29 ppt in 2010, and 1.66 ppt in 2017. Global $c$-$C_4F_8$ emissions started to increase in the 1960s from near
zero to ~1.2 Gg yr$^{-1}$ in the late 1970s to the late 1980s. After this, emissions declined to ~0.8 Gg yr$^{-1}$ in the mid-
1990s to early 2000s. After this emissions again increased, reaching ~2.2 Gg yr$^{-1}$ in 2017. These global emissions are
significantly larger than bottom-up inventory information.
Using the NAME-HB regional inverse method and observations at Gosan station we find that emissions from eastern
Asia rose from ~0.36 Gg yr$^{-1}$ in 2010 to ~0.73 Gg yr$^{-1}$ in 2016 and 2017, representing $31 \pm 4$ % of global emissions,
predominantly from eastern China. Strong $c$-$C_4F_8$ emissions are found from heavily industrialized provinces
south/southwest of Beijing and near the Yangtze River Delta. In contrast, emissions from western Japan, South
Korea, and Taiwan are small, suggesting that their large semiconductor industries are not major $c$-$C_4F_8$ emitters.
Overall, the strong c-$C_4F_8$ emissions in eastern China and their spatial pattern are consistent with emissions from
production of PTFE and other fluoropolymers. A significant fraction of global PTFE production (53–67 % in 2015)





occurs in China and, as indicated by the second largest producer of PTFE in China, the $c$-$C_4F_8$ by-product from the
underlying HCFC-22 pyrolysis process is not recovered or abated, but rather emitted to the atmosphere.
Based on samples collected over the Indian subcontinent in mid-2016, we determine emissions of 0.14 (0.09–0.20)
Gg yr$^{-1}$ $c$-$C_4F_8$ from Northern and Central India (NCI), ~6.8 (4.4–9.7) % of global emissions. Even clearer than in
China, the determined emission map is consistent with emissions from PTFE production.
Using the InTEM regional inverse method and measurements at four western European stations, we only find small
NW European emissions of ~0.02 ± 0.01 Gg yr$^{-1}$ $c$-$C_4F_8$ from 2013–2017 (~1 % of global emissions), in contrast to
an estimate of 14 % of global PTFE production capacity in 2015. The inversion also points to facilities which may
produce PTFE and FEP and/or semiconductor fabrication plants though the picture is less clear.
No obvious correlation between population density and $c$-$C_4F_8$ emissions is found in Eastern Asia, NCI, and NW
Europe, indicating that incineration of waste containing fluoropolymers is not a major source of $c$-$C_4F_8$.
Based on data from two Australian stations and an Inter Species Correlation method, Australian $c$-$C_4F_8$ emissions are
estimated to be small, perhaps ~0.7 % of global $c$-$C_4F_8$ emissions. We find no evidence for $c$-$C_4F_8$ production from
two large aluminum smelters in SE Australia.
Based on a few $c$-$C_4F_8$ pollution events observed at Zeppelin station and a rough FLEXPART analysis, we estimate
that emissions from two Russian facilities known to produce PTFE and halocarbons including $c$-$C_4F_8$ itself could be
~0.24 ± 0.15 Gg yr$^{-1}$. While this could a represent significant fraction of global emissions (possibly ranging from 5 to
26 %), uncertainties are very large.
In summary, for the year 2016, we find global $c$-$C_4F_8$ emissions of 2.06 ± 0.10 Gg yr$^{-1}$, with 0.73 ± 0.12 Gg yr$^{-1}$ from
parts of eastern Asia (36 % of the global total), 0.14 (0.09–0.20) Gg yr$^{-1}$ from Northern and Central India (6.8 %),
~0.02 ± 0.01 Gg yr$^{-1}$ from North Western Europe (~1 %), and ~0.015 Gg yr$^{-1}$ from Australia (~0.7 %).
Current monitoring capabilities of the AGAGE network leave large areas with potential $c$-$C_4F_8$ emission sources un-
or under monitored, e.g. most of the U.S., India, Russia, western China, and eastern Japan where various
semiconductor facilities and fluorochemical and fluoropolymer production plants are located.
While many possible uses and emission sources of $c$-$C_4F_8$ are found in the literature, the start of significant $c$-$C_4F_8$
emissions around the 1960s may well be related to the initial synthesis of PTFE in 1938 with commercial production
of PTFE ("Teflon") by DuPont commencing in 1947 (Gangal and Brothers, 2015) via pyrolysis of HCFC-22 with $c$-
$C_4F_8$ as a by-product/intermediate. It seems unlikely that process control or abatement to minimize $c$-$C_4F_8$ by-
production were in place in the early decades of PTFE production and $c$-$C_4F_8$ by-product was probably emitted to the
atmosphere, explaining the steep increase in global emissions reconstructed here. With the advent of UNFCCC by-
product reporting requirements in the 1990s, concern about climate change and product stewardship, abatement, and
perhaps collection of $c$-$C_4F_8$ for use in the semiconductor industry where it can be easily abated, it is conceivable that
fugitive $c$-$C_4F_8$ in developed countries (UNFCCC Annex 1) overall were reduced, explaining the observed
stabilization and reduction of global emissions in the 1980s and 1990s. Similar efforts to contain and destroy by-
product emissions of fluorocarbons, e.g. HFCs, from the 1980s to the 2000s are documented in the Toxics Release
Inventory (https://www.epa.gov/toxics-release-inventory-tri-program) Program of the US EPA and the European
Pollutant Release and Transfer Register. Concurrently, production of PTFE in China increased rapidly, e.g. from
2000 to 2005 by ~26 % yr$^{-1}$, followed by a slowdown to ~14% yr$^{-1}$ from 2005 to 2015 and perhaps ~8 % yr$^{-1}$ from





2015 onward, reaching an estimated 53–67 % of global production in 2015 (see Tables S2, 3, and 4). Without any
emission reduction requirements, it is conceivable that fugitive emissions of $c$-$C_4F_8$ from PTFE production in China,
and other developing (UNFCC non-Annex 1) countries, today dominate global emissions, in agreement with our
analysis. The 2010 to 2016 rise in rates of eastern Chinese (eastern Asian) $c$-$C_4F_8$ emissions of ~15 % yr$^{-1}$ (~13 % yr$^{-1}$
) determined here are compatible to these PTFE production increase rates of 14 to 8 % yr$^{-1}$ in China. Barring other
developments, we predict that $c$-$C_4F_8$ emissions will continue to rise and that $c$-$C_4F_8$ will become the second most
important PFC emitted to the global atmosphere in terms of $CO_2$-equivalent emissions within a year or two. While
the 2017 radiative forcing of $c$-$C_4F_8$ (~0.52 mW m$^{-2}$) is very small compared to that of $CO_2$, emissions $c$-$C_4F_8$ and
other perfluorinated compounds with similar long lifetimes and high radiative efficiencies essentially permanently
alter the radiative budget of Earth. The fact that significant emissions of ~1.16 Gg yr$^{-1}$ of global emissions (56 %),
exist outside of the monitored regions clearly shows that observational capabilities and reporting requirements need
to be improved to understand global and country wide emissions of PFCs and other synthetic greenhouse gases and
ozone depleting substances.
**7 Author contributions**
JM contributed to archive, firn, and in situ measurements, interpreted the data, and prepared the manuscript with
contributions from all co-authors. CMT provided CSIRO firn model and CSIRO global inversion results and
interpretation. MR provided Bristol global inversion results. LMW provided NAME-HB model runs and emission
estimated for East Asia, DS and ALG provided the same for India. AJM and LMW provided InTEM model runs and
emissions for Europe. AS and NE provided FLEXPART model runs and guided estimation of Russian emissions. DS
and ALG provided the aircraft data from India. CMT, MR, LMW, AJM, DS, ALG, AS, and PJF contributed
significantly to the text. LPS, DJI, TA, JM, PJF, PBK provided and oversaw CSIRO air archive and NH archive
measurements. MKV, SP, SL, M-KP, COJ, LPS, PBK, SOD, PGS, DY, PBK, KMS, OH, BM, CL, JK, JA, MM, SR,
and BY oversaw station operations and provided quality controlled measurement data. PJF provided the estimate of
Australian emissions. CMH provided gravimetric calibration and calibration propagation for the whole AGAGE
network. PKS wrote the software to run all instruments and analyze all measurement data. MKV, BH, CB, VP,
DME, and JS provided firn data and were instrumental in their interpretation. AMcC provided insight into UNFCCC
reporting and bottom-up inventories as well as industrial processes. EM and MC greatly helped with the gathering of
locations of semiconductor facilities. RGP and RFW provided overall project oversight.
**8 Acknowledgments**
Development of the Medusa GC/MS systems, calibrations, and measurements at the Scripps Institution of
Oceanography, La Jolla as well as operations of Trinidad Head, CA were carried out as part of the international
AGAGE research program and supported by the NASA Upper Atmospheric Research Program in the US with grants
NNX07AE89G to MIT, NNX07AF09G and NNX07AE87G to SIO. In the UK, the Department for Business, Energy
& Industrial Strategy (BEIS) provided support through contract 1028/06/2015 to the University of Bristol for Mace
Head, Ireland, for Tacolneston, UK and to the UK Met Office for InTEM analysis. The National Oceanic and



Atmospheric Administration (NOAA) in the US provided support to the University of Bristol for operations at
Ragged Point, Barbados through contract RA-133-R15-CN-0008 and supported operations at Cape Matatula,
American Samoa. Operations in Australia were supported by the Commonwealth Scientific and Industrial Research
Organization (CSIRO), the Bureau of Meteorology (Australia), the Department of Environment and Energy
(Australia), and Refrigerant Reclaim Australia. Operations at Jungfraujoch were supported by the Swiss National
Program HALCLIM (Swiss Federal Office for the Environment, FOEN) and by the International Foundation High
Altitude Research Stations Jungfraujoch and Gornergrat (HFSJG). Operations at Zeppelin were supported by the
Norwegian Environment Agency. Operations at Monte Cimone were supported by the National Research Council of
Italy and the Italian Ministry of Education, University and Research through the Project of National Interest
Nextdata. Operations at Gosan were supported by the National Strategic Project-Fine particle of the National
Research Foundation of Korea (NRF) funded by the Ministry of Science and ICT (MSIT), the Ministry of
Environment (ME), and the Ministry of Health and Welfare(MOHW) (No. NRF-2017M3D8A1092225). Operations
at Shangdianzi were supported by the National Nature Science Foundation of China (41575114). We are indebted to
the staff and scientists at AGAGE and other sites for their continuing contributions to produce high quality
measurements of atmospheric trace gases. Firn air sampling at Law Dome was supported by the Australian Antarctic
Division, Australian Antarctic Science Program, and Australia's Nuclear Science and Technology Organisation. We
acknowledge the members of the firn air sampling teams at South Pole in 2001 and at NEEM in 2008. Firn air
sampling at Summit station was supported through NSF grants ARC-1203779 and ARC-1204084, with airlift
support from the 109th New York Air National guard. We thank E. J. Dlugokencky and the National Oceanic and
Atmospheric Administration (NOAA) Earth System Research Laboratory (ESRL) Global Monitoring Division
(GMD) Carbon Cycle Greenhouse Gases (CCGG) group for measurements which were instrumental for
characterizing the Summit13 firn site. We also thank C.D. Keeling (deceased) and R.F. Keeling (SIO) for air
samples. We thank Dr. T. Saito for helpful discussions. Matthew Rigby was supported in part by advanced research
fellowships from the UK Natural Environment Research Council (NERC, NE/1021365/1). Anita L. Ganesan was
funded under a UK Natural Environment Research Council Independent Research Fellowship (NE/L010992/1). We
acknowledge the contribution of the UK National Environmental Research Council (NERC), the Ministry of Earth
Sciences, Government of India and the Principal Investigators of 'Drivers of the South Asian Monsoon' aircraft
campaign in India. Funding for the measurements used here were made possible by NERC grant NE/I027282/1.



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



**Tables**

**Table 1.** Availability of $c$-$C_4F_8$ in situ, flask, firn, and aircraft air measurements, measurements sites, and instrumentation

| Station/Site | Network | Lat. | Lon. | Medusa no. | Data availability[*] |
|---|---|---|---|---|---|
| Zeppelin (ZEP), Ny-Ålesund, Svalbard | AGAGE | 78.9 | 11.9 | 19 | 09/2010–12/2017 |
| NEEM08 firn, Greenland | – | 77.5 | -51.1 | 9 | Extracted 07/2008 |
| Summit13 firn, Greenland | – | 72.7 | -38.6 | 7 | Extracted 05/2013 |
| Mace Head (MHD), Ireland | AGAGE | 53.3 | -9.9 | 2 | 06/2010–12/2017 |
| Tacolneston (TAC), United Kingdom | UK DECC/AGAGE | 52.5 | 1.1 | 13 | 05/2013–12/2017 |
| Jungfraujoch (JFJ), Switzerland | AGAGE | 46.5 | 8.0 | 12 | 11/2008–12/2017 |
| Monte Cimone (CMN), Italy | AGAGE/ICO-CV | 44.2 | 10.7 | ADS-GC/MS | 05/2013–12/2017 |
| Trinidad Head (THD), USA | AGAGE | 41.0 | -124.1 | 4 | 06/2010–12/2017 |
| Shangdianzi (SDZ), China | AGAGE/CMA | 40.7 | 117.1 | 17 | 05/2010–08/2012, 15/2015–04/2017, 09/2017–12/2017 |
| Gosan (GSN), South Korea | AGAGE/KNU | 33.3 | 126.2 | 10 | 06/2010–09/2016, 04/2017–09/2017, 12/2017–12/2017 |
| La Jolla (SIO), USA | AGAGE | 32.9 | -117.3 | 1 | 11/2009–08/2013, 01/2014–12/2017 |
| NH flasks | SIO & other | 33–46 | -72 – -124 | 7, 1, 9 | 10/1973–04/2016 |
| Aircraft flask samples, India | FAAM/UoB | 9–28 | 72–86 | 21 | 06/2016–07/2016 |
| Ragged Point (RPB), Barbados | AGAGE | 13.2 | -59.4 | 5 | 06/2010–06/2014, 10/2014–12/2017 |
| Cape Matatula (SMO), American Samoa | NOAA/AGAGE | -14.2 | -170.6 | 6 | 08/2010–12/2017 |
| Aspendale (ASA), Australia | AGAGE | -38.0 | 145.1 | 9 | 04–10/2010, 05–07/2011, 05/2015–12/2017 |
| Cape Grim (CGO), Australia | AGAGE | -40.7 | 144.7 | 3 | 09/2010–12/2017 |
| CGAA flasks, Australia | CSIRO/BoM | -40.7 | 144.7 | 9, 7 | 04/1978–12/2010 |
| DSSW20K firn, Antarctica[+] | – | -66.7 | 112.8 | 7 | Extracted 12/1997 |
| SPO01 firn, Antarctica | – | -90.0 | -119 | 9 | Extracted 01/2001 |

[*]Shorter interruptions are excluded.

AGAGE: Advanced Global Atmospheric Gases Experiment (Prinn et al., 2018).

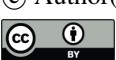

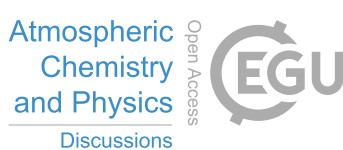

NEEM08: Firn air samples collected in 2008 at the Northern Greenland Eemian Ice Drilling Project, Greenland were collected by the University of Copenhagen,
Denmark, the NEEM consortium, and the Commonwealth Scientific and Industrial Research Organisation (CSIRO) (Buizert et al., 2012).
Summit13: Firn samples collected in 2013 near Summit station, Greenland by the University of Rochester and Oregon State University.
UK DECC: The Tacolneston (TAC) site is part of the UK Deriving Emissions linked to Climate Change network (Stanley et al., 2018).
DSSW20K: Firn samples collected in December 1997 at Dome Summit South West 20 km, Law Dome by CSIRO, the Australian Antarctic Division (AAD), and the
Australian Nuclear Science and Technology Organisation (ANSTO) (see Trudinger et al., 2016 and citations therein).
SPO01: Firn samples collected in 2001 at South Pole, Antarctica, by Bowdoin College, the National Oceanic and Atmospheric Administration (NOAA), the
University of Colorado and the National Science Foundation (NSF) (Aydin et al., 2004; Sowers et al., 2005).
ICO-OV: Measurements at the Italian Climate Observatory "O. Vittori" Monte Cimone (CMN) were performed with a commercial Adsorption-Desorption System
with gas chromatograph and mass spectrometer (ADS-GC/MS) (Maione et al., 2013).
CMA: China Meteorological Administration.
KNU: Kyungpook National University, South Korea.
SIO & other: Most archived northern hemispheric (NH) samples were collected by the Scripps Institution of Oceanography, La Jolla and measured on Medusa 7 .
FAAM/UoB: Air samples over India and the Indian Ocean were taken aboard the UK's FAAM (Facility for Airborne Atmospheric Measurements) BAe-146 research
aircraft and analyzed on Medusa 21 at University of Bristol (UoB) (Say et al., 2019).
CGAA: Cape Grim Air Archive samples were collected by the CSIRO Oceans and Atmosphere and the Bureau of Meteorology (BoM), Australia predominantly
measured on the Aspendale Medusa 9 at CSIRO (Langenfelds et al., 2014).














**Table 2.** Regional c-C$_4$F$_8$ emissions derived for eastern Asia from Gosan measurements (NAME-HB inversion) and comparison to global emissions (Gg yr$^{-1}$, kt yr$^{-1}$)

| | Eastern China[#] | Western Japan[#] | South Korea | North Korea | Taiwan[#] | Σ eastern Asia | Global[+] | Global - Σ eastern Asia |
|---|---|---|---|---|---|---|---|---|
| 2010 | 0.30 ± 0.07 | 0.02 ± 0.01 | 0.019 ± 0.008 | 0.008 ± 0.004 | 0.008 ± 0.005 | 0.36 ± 0.07 | 1.40 ± 0.11 | 1.04 ± 0.13 |
| 2011 | 0.35 ± 0.07 | 0.02 ± 0.01 | 0.016 ± 0.007 | 0.006 ± 0.003 | 0.007 ± 0.005 | 0.41 ± 0.07 | 1.52 ± 0.10 | 1.12 ± 0.12 |
| 2012 | 0.41 ± 0.06 | 0.02 ± 0.01 | 0.009 ± 0.005 | 0.004 ± 0.002 | 0.010 ± 0.008 | 0.45 ± 0.06 | 1.61 ± 0.08 | 1.16 ± 0.10 |
| 2013 | 0.46 ± 0.09 | 0.02 ± 0.01 | 0.017 ± 0.007 | 0.007 ± 0.004 | 0.008 ± 0.005 | 0.51 ± 0.09 | 1.67 ± 0.09 | 1.15 ± 0.13 |
| 2014 | 0.54 ± 0.06 | 0.03 ± 0.01 | 0.039 ± 0.009 | 0.009 ± 0.004 | 0.009 ± 0.006 | 0.62 ± 0.06 | 1.76 ± 0.09 | 1.14 ± 0.11 |
| 2015 | 0.59 ± 0.09 | 0.02 ± 0.01 | 0.041 ± 0.010 | 0.011 ± 0.005 | 0.011 ± 0.009 | 0.68 ± 0.09 | 1.88 ± 0.10 | 1.21 ± 0.13 |
| 2016 | 0.67 ± 0.12 | 0.02 ± 0.01 | 0.022 ± 0.010 | 0.009 ± 0.005 | 0.009 ± 0.006 | 0.73 ± 0.12 | 2.06 ± 0.10 | 1.33 ± 0.16 |
| 2017 | 0.68 ± 0.13 | 0.02 ± 0.01 | 0.014 ± 0.011 | 0.006 ± 0.005 | 0.010 ± 0.009 | 0.73 ± 0.13 | 2.20 ± 0.11 | 1.47 ± 0.17 |
| a priori[*] | China 0.42 ± 0.05 | Japan 0.09 ± 0.01 | South Korea 0.032 ± 0.002 | North Korea 0.010 ± 0.001 | Taiwan 0.009 ± 0.001 | Sum 0.56 ± 0.05 | | |
| a priori[*] | Eastern China 0.185 | Western Japan 0.0294 | | | | | | |

[+]Global emissions are the average of the emissions determined by the CSIRO and the Bristol inversion in this work.

[#]Eastern China contains the provinces Anhui, Beijing, Hebei, Henan, Jiangsu, Liaoning, Shandong, Shanghai, Shanxi, Tianjin and Zhejiang. Western Japan contains the prefectures Chugoku, Kansai, Shikoku and Okawa and Kyushu. Due to the lower sensitivities of the inversion in western China, eastern Japan, and parts of Taiwan, where potential source industries are located, we cannot exclude further emissions in these regions and therefore total emissions are probably larger.

[*]Saito et al. (2010) emission estimates based on atmospheric measurements from November 2007 to September 2009 were used as a priori information and were spread for each country uniformly over the area of each country. The resulting a priori estimates for eastern China and western Japan are additionally listed for comparison with the inversion results for these regions.

Gosan measurements started in June 2010 with most complete coverages from 2011 to 2015.




**Figures**




**Figure 1**. $c$-$C_4F_8$ mole fractions reconstructed from the late 1970s to 2018 from archived air samples and in situ
measurements in both hemispheres. Cape Grim Air Archive (CGAA) and archived NH air samples are shown with
symbols in shades of green and blue, respectively, reflecting different data subsets. For recent years, in situ
measurements are shown as pollution removed monthly means for extra-tropical stations in the NH (MHD in light
blue, THD in orange, SIO in darker blue, JFJ in grey) and in the SH (CGO in lighter green, ASA in pale green).
Shown are the final data after an iterative filtering process described in the main text. The final suggested fits are
shown as bold light green (SH) and bold light blue (NH) polynomial fits. Results for the tropical stations, RPB and
SMO, the Asian stations, GSN and SDZ, and the Arctic station, ZEP, are omitted here for clarity.



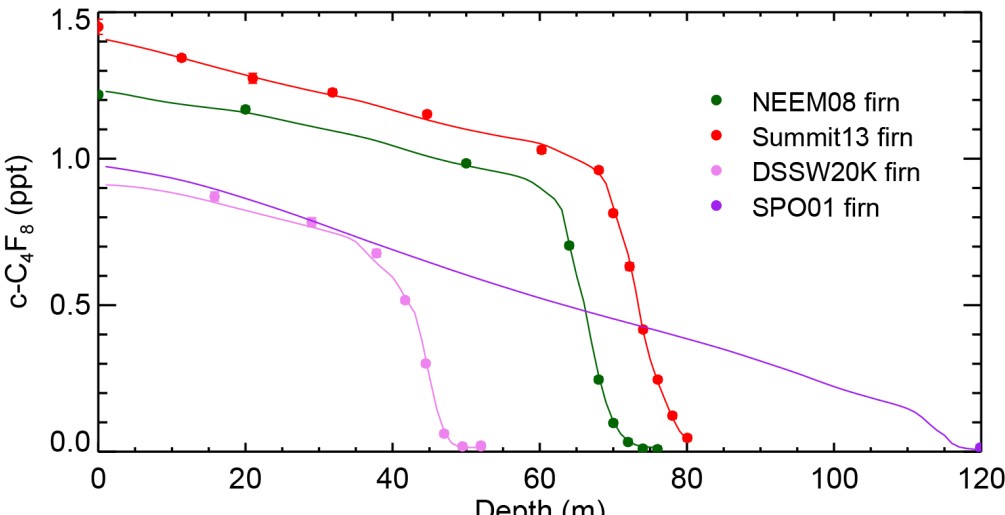


**Figure 2**. Depth profile of $c$-C$_4$F$_8$ measured dry-air molar mole fractions (parts per trillion, ppt) in air extracted from polar firn at NEEM08 (Northern Greenland, dark green) and Summit13 (Greenland, red) in the NH and DSSW20K (Eastern Antarctica, pink) and SPO01 (South Pole, purple) in the SH, together with the simulated depth profiles for each site (dark green, red, pink, and purple lines) that correspond to the emissions inferred by the CSIRO inversion. The modelled depth profiles for each site (solid lines) are based on the inversion of measurements from all firn sites, archive, and in situ data. Measurement precisions (1σ) are shown as error bars and are generally smaller than the plotting symbol.



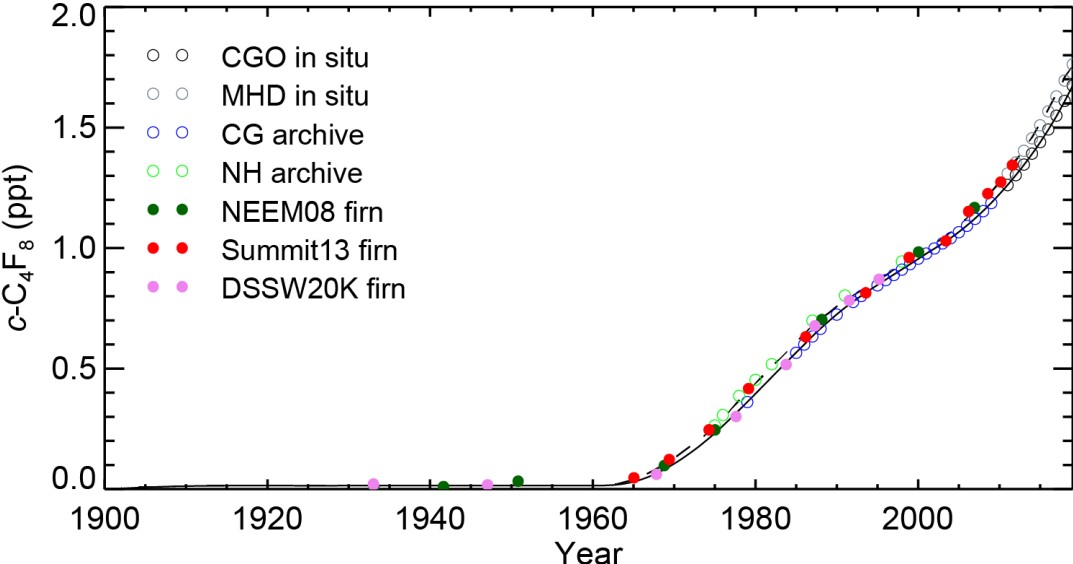




**Figure 3**. Historic atmospheric $c$-$C_4F_8$ mole fractions reconstructed for the extra-tropical Northern and Southern
Hemispheres from air extracted from polar firn (full circles, NEEM08 in dark green, Summit13 in red, DSSW20K in
pink, against mean or effective ages; SPO01 with mean age of ~1890 is not shown), annual values from spline fits to
Cape Grim Air Archive (CG archive, open blue circles) and in situ measurements at Cape Grim (CGO, open black
circles), archived air samples (NH archive, open green circles) and in situ measurements at Mace Head (MHD, open
grey circles). Also shown are reconstructed abundances based on optimized emissions determined by the CSIRO
inversion for the extra-tropical SH (black line) and NH (dashed black line).

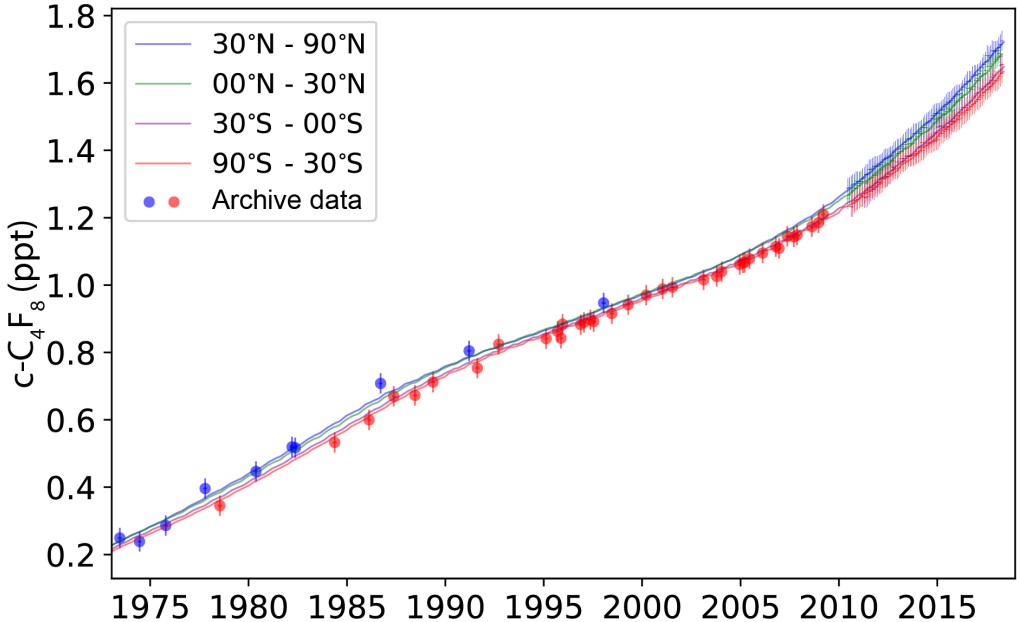


**Figure 4**. Historic $c$-$C_4F_8$ mole fractions from archive samples in both hemispheres (filled circles) and pollution free
monthly mean in situ data from AGAGE background sites (MHD and THD in blue, RPB in green, SMO in purple
and CGO in green, vertical bars, bar size represents variability of monthly means) are shown together with the
Bristol inversion results for the four latitudinal bands represented by these background sites (30° N–90° N, 0° N–30°
N, 0° S–30° S and 30° S–90° S, solid lines of same color).












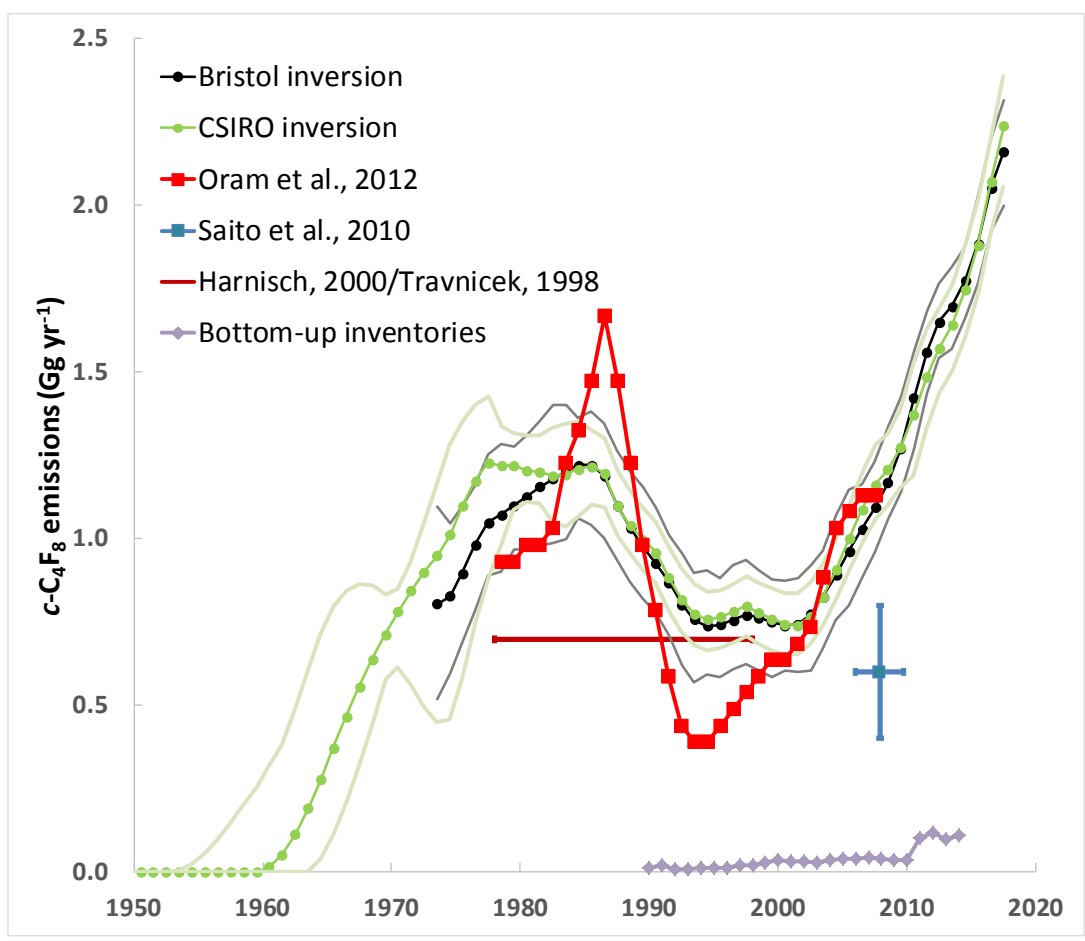


**Figure 5**. Global $c$-$C_4F_8$ emissions reconstructed by the CSIRO inversion (green dots and line, light green 2 σ
uncertainty bands) from 1950 and by the Bristol inversion (black dots and line, grey 1 σ uncertainty bands) from the
early 1970s to present. In situ and archive data are used in both inversions, while firn air data are only used in the
CSIRO inversion. Emission estimates by Oram et a., 2012 (red), Saito et al., 2010 (blue), Harnisch, 2000/Travnicek,
1998 (brown) and from available bottom-up inventory information (grey) are shown for comparison.



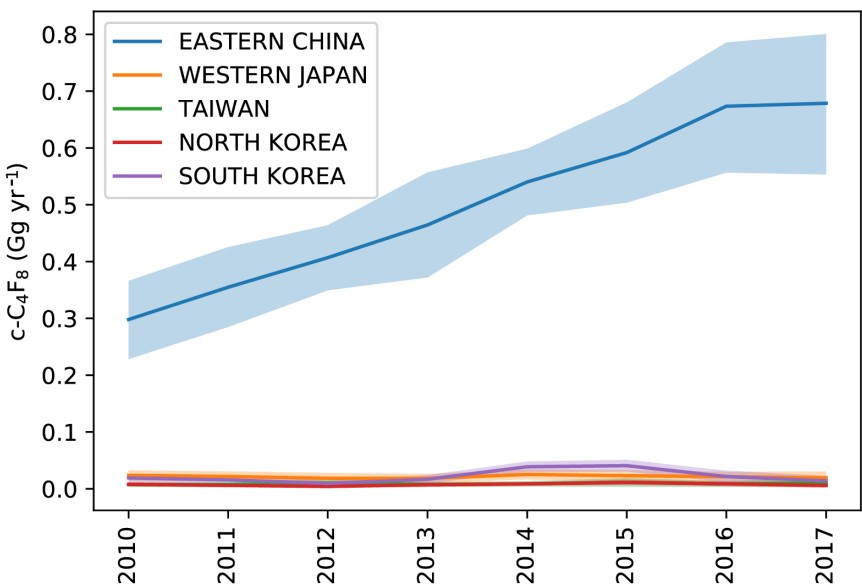

1294

**Figure 6**. $c$-$C_4F_8$ emission in eastern Asia as determined by the NAME-HB regional inversion of measurements at
the Gosan station, Jeju Island, South Korea are dominated by emissions from eastern China (blue), followed by
emissions from western Japan (orange). Emissions from South Korea (violet) are much smaller, but show a small
maximum in 2014 and 2015. Emissions from Taiwan (green) and North Korea (red) also small. Shadings represent
uncertainty bands of emissions.





















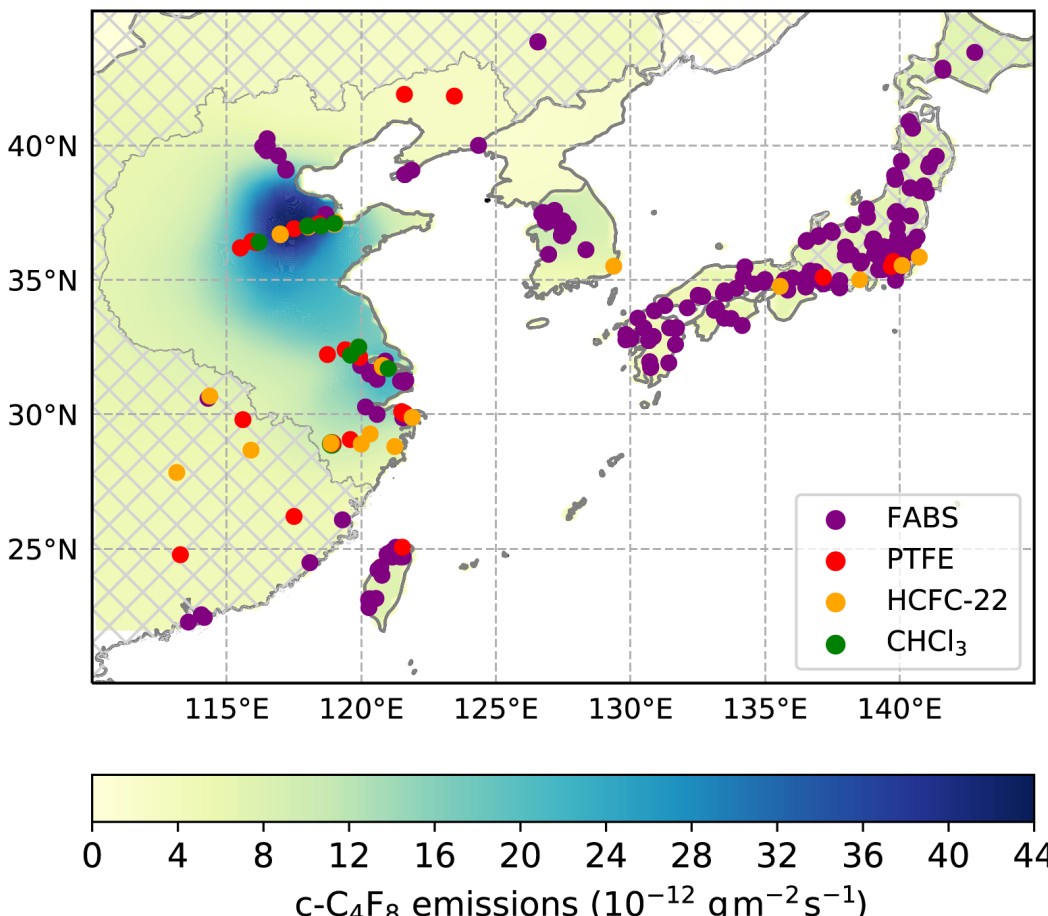

**Figure 7**. Mean $c$-$C_4F_8$ emission strength (shades of green and blue, $10^{-12}$ g m$^{-2}$ s$^{-1}$) in eastern Asia from 2010 to 2017 determined by the NAME-HB inversion from measurements at the Gosan station, Jeju Island, South Korea. The hatching indicates areas for which emissions are not reported due to relatively low sensitivities of the inversion. Emissions predominantly occur in the densely industrialized Shandong, Tianjin and parts of Henan and Hebei provinces south/southwest of Beijing as well as in Shanghai and neighboring provinces Jiangsu (to the north), Anhui (to the west) and Zhejiang (to the south) of the Yangtze River Delta region. Shown are industries with potential $c$-$C_4F_8$ emissions: Semiconductor fabrication plants (FABS, purple dots, en.wikipedia.org/wiki/List_of_ semiconductor_fabrication_plants, www.10stripe.com/featured/map/semiconductor-fabs.php and other sources) and TFE/HFP/PTFE/FEP production facilities (PTFE, red dots, www.qianzhan.com/analyst/detail/220/170629-c33a2ca7.html and other sources). HCFC-22 (orange dots) and chloroform (CHCl$_3$, green dots) production facilities are shown as the TFE and HFP monomers needed to produce PTFE and FEP fluoropolymers are produced via pyrolysis of HCFC-22 and $c$-$C_4F_8$ is an intermediate/by-product in this process, while HCFC-22 is manufactured from CHCl$_3$.




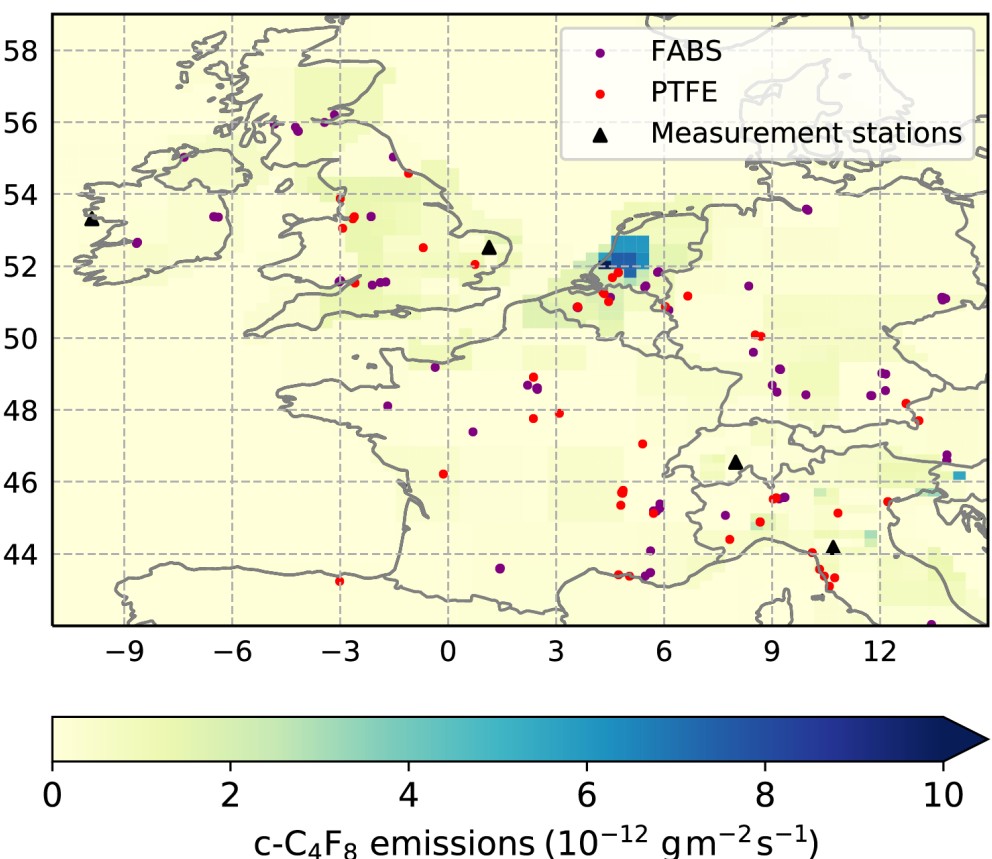

1332

**Figure 8**. Mean $c$-$C_4F_8$ emission strength (shades of green and blue, $10^{-12}$ g m$^{-2}$ s$^{-1}$) in North Western Europe (42° N to 59° N and -11° E to 15° E) from 2013–2017 determined by the InTEM inversion from measurements at four sites (Mace Head, Ireland, Tacolneston, United Kingdom, Jungfraujoch, Switzerland, and Monte Cimone, Italy, black triangles). Also shown are potential industrial emitters of $c$-$C_4F_8$. Locations of potential TFE/HFP/PTFE/FEP production facilities (red dots) are based on company websites (3M, Chemours, Daikin, DuPont, Saint-Gobain, and Solvay) and are much less certain than the corresponding location information for eastern Asia. Also shown are semiconductor fabrication plants (purple dots, en.wikipedia.org/wiki/List_of_semiconductor_fabrication_plants, www.10stripe.com/featured/map/semiconductor-fabs.php, and other sources).








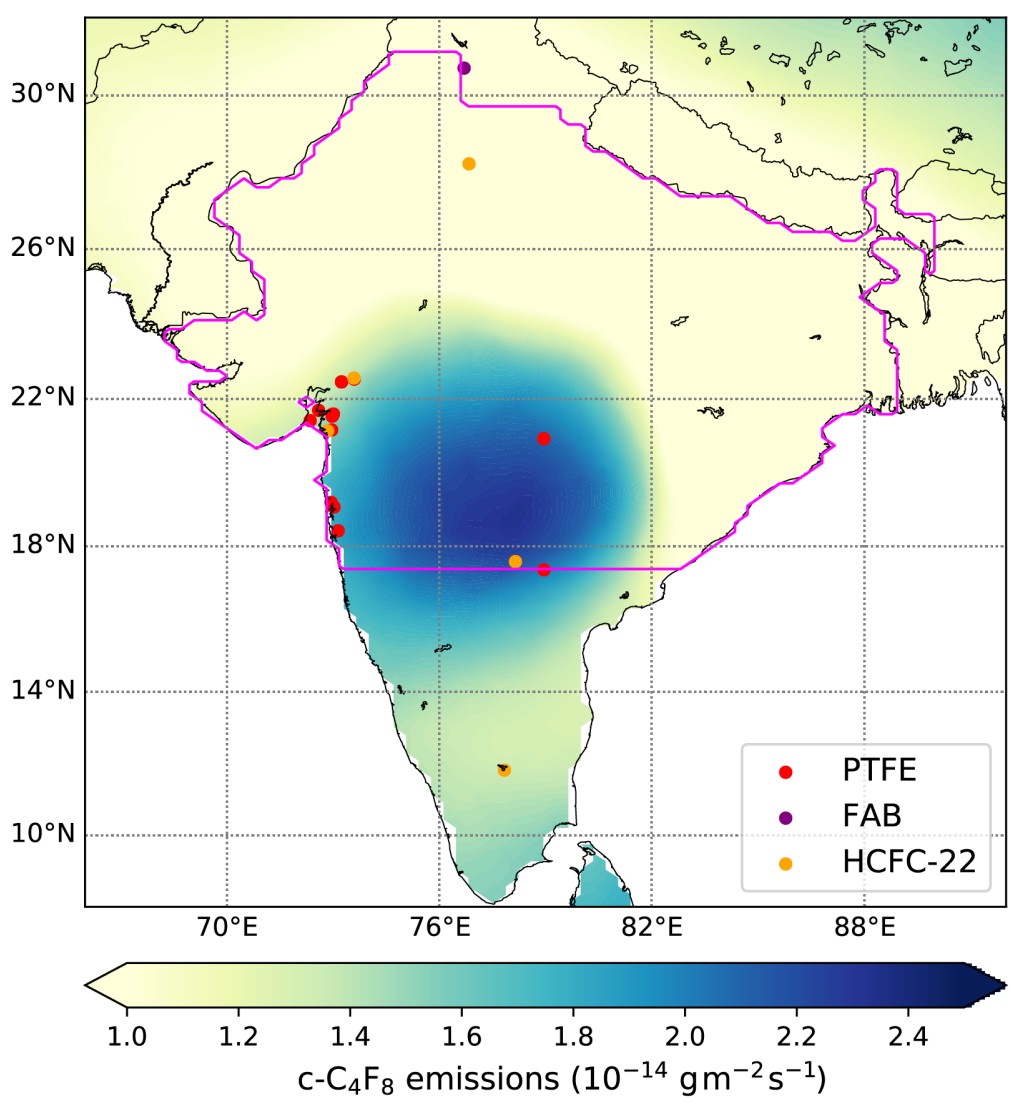


**Figure 9**. Mean $c$-C$_4$F$_8$ emission strength (shades of green and blue, $10^{-14}$ g m$^{-2}$ s$^{-1}$) over the Indian subcontinent for June and July 2016 determined by the NAME-HB inversion based on air samples taken on-board UK's FAAM (Facility for Airborne Atmospheric Measurements) BAe-146 research aircraft. Also shown are the locations of one semiconductor fabrication plant (FAB, purple dot) and several potential PTFE/FEP production facilities (PTFE, red dots, Solvay/CYTEC, Hindustan Fluorocarbons, and Gujarat Fluorochemicals facilities) as potential $c$-C$_4$F$_8$ sources. HCFC-22 (orange dots) production facilities are also shown as the TFE and HFP monomers needed to produce PTFE and FEP fluoropolymers are produced via pyrolysis of HCFC-22 and $c$-C$_4$F$_8$ is an intermediate/by-product in this process. The outline of the Northern and Central India (NCI) model domain is shown as a pink line.

1357