# Peer review of "Perfluorocyclobutane (PFC-318, c-C4F8) in the global"

_Atmospheric Chemistry and Physics, 2019_

## Referee Comment (RC1) · Anonymous Referee #1 · 28 May 2019

The authors have brought together an impressive data set for atmospheric observations and modelling of PFC-318, which is global, long-term and consistent. The manuscript is well written but perhaps lacking succinctness a bit, which could be helped by moving some rather technical aspects that are not essential for the narrative to the supplement. A general concern is that the authors focus very much on their hypothesis of PTFE production as the main source of PFC-318 to the atmosphere. Other sources are barely mentioned let alone discussed, whereas the observations in my opinion point to a much more complex picture of emission sources (including unknowns). This should be given some more consideration. Other points are listed below.

L119 I don't think a personal communication can be counted as evidence

L169-178 Have the authors ascertained that their calibration system has a linear re-

sponse behaviour over a relevant mole fraction range? How was the calibration scale uncertainty estimated?

L187 "perhaps slightly better"?

L210 "perhaps"?

L214-249 It is commendable that the authors have carried out these tests. However, the high number of statistical outliers is worrying and casts some doubt on the derived long-term trends, in particular the early parts. Adding uncertainty ranges to the fits based on a) the samples that were included but showed discrepancies between Medusas and b) the sparsity of the measurements, might help here. In addition, I recommend moving this rather technical paragraph to the supplement.

L225 "eigth"

L267 "a"

L513 How high is the interhemispheric gradient and how has it evolved over time? This might e.g. reveal information on changes in emission latitudes. There is a lot of space in Figures 1 and 4 to show this.

L523 Define "good agreement". There are no uncertainty ranges given for the two estimates in Figure S7.

L537-538 Again, are these discrepancies within uncertainties?

L555 GWP-100?

L571-574 Given that the largest emissions appear to occur near the sea, is there scope for some emissions being related to ships or submissions? What fraction of emissions did the model initially assign to have occurred over the ocean?

L590-591 This appears to be in disagreement with the statement in L576-578.

L603 FABS?

L631-638 Please add information such as measurement precisions, observed mole fraction ranges, ions used for identification and quantification, etc. on the HFP measurements to the manuscript. Please provide quantitative evidence instead of "associated with" and "virtually absent".

L639-641 Consistent with emissions from many of these facilities, but clearly not all (as stated in L610-612). Given the problems with associating these sources can the authors confirm that the ratios between m/z 131 and 101 during pollution events were consistent with those observed in clean air? This would help to rule out interferences during pollution events.

L661 How much smaller?

L697 That is a very optimistic way of looking at that Figure.

L706-707 This is not very clear from the Figure, which is rather indicating an unknown source.

L713 What is the main purpose of this direct c-C4F8 production?

L729 Which ones did PFC-318 correlate best with (also for other pollution events in Asia etc)?

L740-741 How much larger?

L1236 Figure 1 is mostly demonstrating quality assurance purposes and one cannot see most station data anyway as it is on top of each other. As the long-term trend is shown again in Figures 3 and 4 I suggest moving it to the supplement.

L1256 Is it necessary to show years from 1900 if the first data point is after 1930? What is the uncertainty of the calculated effective ages?

Supplement

Figure S1 The caption is actually an entire section and should perhaps have its own

[Figure]

heading.

Figure S3 There is quite some uncertainty in the 1960s and 70s. Has this been reflected in the emission uncertainties?

Figure S7 Please also show the published observational data set of Saito et al.

Table S3 RoW is not explained and web pages should be cited with the date on which they were accessed.

---

## Referee Comment (RC2) · Anonymous Referee #2 · 29 May 2019

The manuscript entitled "Perfluorocyclobutane (PFC-318, c-C4F8) in the global atmosphere" by Mühle et al. has been evaluated by this reviewer. The paper presents a substantial piece of measurement and modeling work on the atmospheric abundance and emission rates of perfluorobutane. The authors have developed an independent gravimetric c-C4F8 calibration scale and characterized the abundance of c-C4F8 with high precision in both hemispheres in order to determine historical emissions (archived samples) and recent global emissions. Using inversion modeling techniques, regional emission patterns (and pollution events) are investigated in detail, revealing that major c-C4F8 sources are found in heavily industrialized provinces of China (and perhaps Russia), due to the production of PTFE and other fluoropolymers. They predict c-C4F8 emissions will continue to rise and that c-C4F8 will become the second most important

[Figure]

PFC emitted to the atmosphere in terms of CO2 equivalent emissions.

General Comments: The manuscript is a pleasure to read, has very few technical errors, and presents and an impressive amount of interesting data. The authors have done a commendable job to present a succinct and encompassing description of the methods and the results. The conclusions follow elegantly from the data presented and form a compelling narrative, especially considering the magnitude of difference in the potential emissions sources involved. I have only few scientific comments/questions. Those are listed below here, followed by technical (suggested) corrections:

Specific Comments: L55: ".... explaining the increase in emissions." Presumably the authors here refer to the early/pre 1980's? L65: "Significant emissions" must be inferred significant emissions? L115: What is "aerolyzed foods"? Please explain (very briefly), or used more common term. L262: Please explain what is meant by "above bubble close-off". L.396-397: "it was assumed that emissions were constant from year to year". This seems confusing to me. Perhaps I'm not understanding this inversion correctly. I can see that the emissions would be assumed constant during the year, but why from year to year? How does this work? L440: "We do not report emission estimates outside of eastern Asia due to large posterior uncertainties but include them assisted with determination of the boundary conditions". I do not understand this approach. Please clarify and explain. L538: Please explain why not incorporating the firn data has this impact on the emissions estimates. L547: How can the mole fractions of this very unreactive compound change in the tanks? L553: C2F6 is here listed as a minor PFC, however in L122, it was a major. Which are the majors and the minors? L558: To make clear what we are talking about, I suggest inserting "from a climate forcing standpoint" before "will become the second most important PFC...".

"Technical" Comments: L42-43: The propagated uncertainties on the emissions should be given in the abstract. L61: " in agreement with our analysis". This seems like an obvious statement. Suggest deleting. L102: "Recently there is also further evidence...". This sentence begins awkwardly – suggest rewording it. L249: "Fig.1." Other places

in the manuscript "Figure" is used. Check for consistency. L302: Perhaps "default" is a better word in place of "definition". L311: Move the definition of 1t =0001Gg to the introduction paragraph, where Gg is first used. L316: replace "similar to" with "analogous to what has been observed for". L641: Insert "occurring" before "...in China...) Line 689-691: These two sentences belong more appropriately in sections 5.35 and 5.3.6.

Figures: Figures 1 is nicely formatted, but the formatting is inconsistent with that applied in Figures 2-4. Moreover Figure 5 has a completely different formatting style. This figure formatting ought to be "harmonized".

Figure 1 caption: What are the error bars?

––––––––––––––––––––––––––––

---

## Author Response (AR1)

Mühle et al., Perfluorocyclobutane (PFC-318, c-C4F8) in the global atmosphere, Atmospheric Chemistry and Physics, acp-2019-267

- 4 Authors' response to reviews:
- 5

3

Below we repeat the suggestions from both reviewers in italic and add our replies in bold. If we quote sentences from
the manuscripts, modified parts will be bold, while unmodified parts will not be bold. We thank both reviewers for
their overall positive reviews and helpful suggestions.

9

**10 **Reviewer 1:**

The authors have brought together an impressive data set for atmospheric observations and modelling of PFC-318, which is global, long-term and consistent. The manuscript is well written but perhaps lacking succinctness a bit, which could be helped by moving some rather technical aspects that are not essential for the narrative to the supplement. A general concern is that the authors focus very much on their hypothesis of PTFE production as the main source of PFC-318 to the atmosphere. Other sources are barely mentioned let alone discussed, whereas the observations in my opinion point to a much more complex picture of emission sources (including unknowns). This should be given some more consideration. Other points are listed below.

18 We thank the reviewer for the overall positive assessment of our work. We agree that more technical aspect 19 can be moved in the Appendix. In our detailed replies and our revisions to the manuscript we now point out 20 more strongly the limitations of the inversion. Foremost, the further away emissions occur, the more likely the 21 regional inversion method will allocate these emissions to a general diffuse region, rather than identify 22 individual  $c-C_4F_8$  point sources. For the Indian subcontinent, the limited number of samples taken onboard 23 the aircraft contributes further to the problem. This needs to be kept in mind when interpreting the inversion 24 results for East Asia and India. We also now emphasize that we cannot categorically exclude an unknown 25 industrial source (Abstract, Section 5.3.5, Summary and Conclusion) and changed or added several statements (Abstract, Sections 5.3.1 and 5.3.5, and Summary and Conclusion), but the data and the inversion 26 27 results are consistent with the hypothesis that production of TFE/HFP/PTFE/FEP and other fluorochemicals, 28 both historically in developed countries and today in developing countries are likely the main source for c-29  $C_4F_8$ . c- $C_4F_8$  is a by-product of the production of the needed TFE and HFP monomers via the pyrolysis of 30 HCFC-22, industry experts confirm the practice to vent  $c-C_4F_8$  from this process into the atmosphere 31 (historically in developed countries and currently in China), emissions are not correlated with population 32 density, and the semi-conductor industries in South Korea, Japan, Taiwan, and Europe do not emit significant 33 amounts of c-C4F8. We hope that our added explanations and revisions address the concerns of the reviewer.

34

**35 L119 I don't think a personal communication can be counted as evidence**

36 In hindsight, the evidence was weak and it seems that  $c-C_4F_8$  was not used significantly in this application.

37 Lacking other references, we removed this part of the sentence from the Introduction. The evidence for use as

- 38 geohydrological tracer is also rather weak and we chose to add "perhaps used as a geohydrological tracer".
  - 1

- 39
- 40 *L169-178 Have the authors ascertained that their calibration system has a linear response behaviour over a relevant*41 *mole fraction range? How was the calibration scale uncertainty estimated?*
- 42 As explained in the last paragraph of Section 2.1, the linearity of the analytical system was assessed "with a 43 series of diluted air samples (parent tank at 1.252 ppt, dilutions from 100 % to 6.25 %, Ivy et al., 2012) and a series of different volumes of a working standard (parent tank at 1.60 ppt, sample volumes from 200 % to 5 % of usual 2 L 44 45 volume). A small deviation from linearity was observed for the most diluted samples and the smallest volumes 46 probably due to a memory or blank of ~0.014 ppt on Medusa 9 was corrected for. Medusa 7 showed an effect of 47  $\sim 0.008$  ppt, but as this was just below the detection limits and within the typical precisions, we chose not to correct 48 for this.". We hope that the reviewer agrees that these tests are sufficient to establish linearity. 49 The calibration scale uncertainty was estimated conservatively based on the purity of the reagent, the 50 reproducibility of the dilution technique to prepare the mixtures, measurement precisions, and propagation

51 uncertainties as outlined in Prinn et al., 2000, 2001, and 2018 which we added at the end of the second 52 paragraph of Section 2.1.

53

**54 *L187 "perhaps slightly better"?**

55 We tried to express that the 5975 series MSDs showed slightly better detection limits than 5973 series MSDs. 56 As the range of ~0.01-0.03 ppt includes detection limits estimated for both 5973 and 5975 series MSDs, we 57 changed the sentence to "Detection limits .... were ~0.01–0.03 ppt for both types of MSDs.".

58

59 L210 "perhaps"?

We agree with the reviewer and removed "perhaps". It was redundant as the slight uncertainty in the estimated effect of ~0.008 ppt is already reflected in the approximate sign (~).

62

L214-249 It is commendable that the authors have carried out these tests. However, the high number of statistical outliers is worrying and casts some doubt on the derived longterm trends, in particular the early parts. Adding uncertainty ranges to the fits based on a) the samples that were included but showed discrepancies between Medusas and b) the sparsity of the measurements, might help here. In addition, I recommend moving this rather technical

- 67 *paragraph to the supplement.*
- We understand the reviewer's concern, which perhaps arises in part from the lengthy description of the tests 68 69 we performed to verify that measurements at CSIRO and SIO agree, intermingled with the description of the 70 filtering of the actual air archive measurements. We have now moved the discussion of the tests performed at 71 CSIRO and SIO as well as some of the other details into the Supplement (at the beginning). Following the 72 reviewer's suggestion, this considerably shortened Section 2.2 and streamlined the description of the archive 73 data filtering. As suggested by the reviewer, we have also added the 95% confidence bands to the fits in Fig. 1. 74 It is correct that the "early part" of the record, from the mid-1970s to the late 1970s, is more uncertain as no Cape Grim Air Archive (CGAA) data were available, but we would like to point out that the reconstruction of 75 76 the "early part" of the record, from the late 1970s until in-situ data became available, is dominated by the

77 frequent and well behaved CGAA data. The less frequent, filtered NH tank data play a rather unimportant 78 role. In the fits in Fig 1, this was achieved as described in the text by guiding the NH fit with the CGAA data. 79 Note that, in the CSIRO inversion, the same effect was essentially achieved by using larger data uncertainties 80 for the NH data, so that the inversion fits more closely to the SH data. When the NH tank data are left out of 81 the CSIRO inversion, the reconstructed mixing ratios and emissions do not change significantly (see Fig. S3a 82 for the emissions, and we now include the sensitivity test results for the reconstructed mixing ratios as well in 83 a modified Fig. S3b). Moreover, it should be pointed out that most of the filtered NH tanks were filled in 2003 84 and later, typically many tanks on one or two days in a given year, which would add little information to the 85 reconstruction given the onset of in-situ data at multiple stations in 2011 and the high quality of the CGAA 86 data used to guide the filtering. We added this explanation to the revised text in Section 2.2. 87 88 L225 "eigth" 89 We fixed this typo to "eighth" and the text has been moved to the beginning of the Supplement. 90 91 L267 "a" 92 We removed this orphan "a" from Section 2.3. 93 94 L513 How high is the interhemispheric gradient and how has it evolved over time? This might e.g. reveal 95 information on changes in emission latitudes. There is a lot of space in Figures 1 and 4 to show this. 96 We believe that the IH gradient before the onset of in-situ data is too dependent on the more scattered and 97 uncertain NH archive data to draw defendable conclusions. From 2011 to 2017, when full in-situ datasets are available, the IH gradient increased from ~0.05 ppt to ~0.09 ppt, in line with increasing predominantly NH 98 99 emissions. We added a new inset to Fig. 1 to show this, modified the caption, and added corresponding text to 100 Section 5.1. Unfortunately, we do not believe that we can draw conclusions about changes in emission latitudes 101 from the IH gradient with the 12-box atmospheric model and annually-repeating transport parameters. 102 103 L523 Define "good agreement". There are no uncertainty ranges given for the two estimates in Figure S7.

- We felt that the old Fig. S7 would be too busy if we had included the uncertainty bands, but we agree with the reviewer that we need to show them. Therefore, we created a new Fig. S7 which shows the reconstructed mole
- 106 fractions by the two inversions including  $2\sigma$  uncertainty bands, demonstrating the good agreement of the two
- 107 inversions. We reference this new Fig. S7 in Section 5.1. We also included the uncertainties of the mean ages
- 108 (before 1965) and effective ages (after 1965) for the firn samples in the new Fig. S7 with respect to the question
- 109 of reviewer #1 about L1256. (Note, the old Fig. S7 is now new Fig. S8 and so forth. Due to the insertion of new
- 110 Fig. S12, old Fig. S11 is now Fig. S13 and old. Fig. S12 is now new Fig. S14.)
- 111

**112 L537-538 Again, are these discrepancies within uncertainties?**

- 113 Yes, the discrepancies in global emissions are within uncertainties as can be seen in Fig. 5. Emissions prior to
- 114 **1980** rely on archive data that are predominantly NH, relatively sparse and of poor quality compared to later

115 decades (see Sec 2.2), and, in the case of the CSIRO inversion, firn data that comprises atmospheric air

spanning a range of ages of typically about 40 years. To make this more clear to the reader and to also follow

117 reviewer #1's comment on the same sentence, we modified this sentence at the beginning of Section 5.2 to "The

118 Bristol inversion initially reconstructs lower emissions, but the differences are within the estimated uncertainties

- 119 for the reconstructed histories (see Fig. 5)."
- 120

121 L555 GWP-100?

122 The reviewer is correct. We clarified the text at the end of Section 5.2 accordingly. The use of  $GWP_{100}$  was 123 actually specified in the caption of old Fig. S8/new Fig. S9, but now we also added the  $GWP_{100}$  for each 124 compound as well as a citation.

125

L571-574 Given that the largest emissions appear to occur near the sea, is there scope for some emissions being related to ships or submissions? What fraction of emissions did the model initially assign to have occurred over the ocean?

We are wondering if there is a misunderstanding here. Old Fig. S10/new Fig. S11 shows the cumulative footprint map for 2010-2017 for the Gosan station. This is the sensitivity to potential emissions from each area of the grid box on the map, not the emission strength at any given grid box. The sensitivity to potential emissions is related to how often air originates from a certain grid box according to meteorological models. It reflects how much information about distant sources is collected at the receptor site (in this case Gosan station). The model assigns no emissions over sea a priori, and the inversion does not allow emissions to be placed

there, see Fig. 7. The inversion actually infers a spatially resolved scaling of the a priori emissions field, therefore by setting 0 emissions then no matter how it is scaled, the resultant emissions will always be 0 over sea.

139

140 L590-591 This appears to be in disagreement with the statement in L576-578.

141 We actually disagree, but our wording in Section 5.3.1 was probably not clear. On L590-591 we compare a) 142 the combined regional emissions in East Asia with our global emissions estimate and b) the Eastern Chinese 143 emissions with our global emissions estimate. On L576-578, however, we compare Eastern Chinese emissions 144 determined by our inversion with the a-priori emissions for Eastern China estimated from Saito et al. 145 Therefore, the statements are not in disagreement. To make this clearer, we have changed the sentence near the end of the first paragraph of Section 5.3.1 to "The a priori emissions for eastern China of  $0.185 \text{ Gg yr}^{-1}$  are 146 based on the Saito et al. (2010) estimate for all of China for November 2007 to September 2009, but the inversion 147 148 suggests emissions that are ~62 % higher in 2010 and more than triple in 2017.".

149

**150 *L603 FABS?**

151 We thank the reviewer for catching that we did not define the FABS at the first appearance in the text. We

152 now defined it as "semiconductor fabrication plants (FABS)". It had been defined in the caption for Figure 7.

153

- L631-638 Please add information such as measurement precisions, observed mole fraction ranges, ions used for identification and quantification, etc. on the HFP measurements to the manuscript. Please provide quantitative evidence instead of "associated with" and "virtually absent".
- 157 We apologize for not including more details on the HFP measurements. We now added the new Fig. S12 with 158 a detailed caption. We also modified the main text at the end of Section 5.3.1 and refer to this new Figure. 159 HFP is measured on m/z 131 and 150. On the Porabond O column it elutes after HFC-125 and before CFC-160 115. We confirmed the identify of HFP with a spike of ~10 ppt HFP (87,422 area counts) measured at SIO. 161 The working standard used at that time had a small HFP peak equivalent to ~0.03-0.04 ppt (270-380 area 162 counts), while ambient air samples contained ~0.01 ppt HFP (98-123 area counts), just around the estimated 163 detection limit of ~0.01 ppt (3 times baseline noise). The small abundance of HFP in the working standard led 164 to poor precisions of ~20%. From Nov. 2018 until present, ambient air measurements at SIO typically showed 165 0-0.5 times (0-150 area counts) the response of the working standard used, reaching at most 2.5 times, 166 indicating continuing miniscule ambient mixing ratios. HFP measurements at Aspendale (ASA) have not been 167 calibrated, but the peak responses in ambient air sampled from Feb. 2017 until present were almost always 168 small (ranging from 0 - 300 area counts), indicating similarly small ambient mixing ratios as at SIO. Only 169 occasional small pollution events have been observed at ASA as discussed in Section 5.3.3. HFP measurements 170 at Gosan and Shangdianzi (SDZ) were not calibrated, but several working standards showed significant peak 171 responses (up to 2,500 and 4,000 area counts, respectively). From Aug. 2018 until present,  $c-C_4F_8$  pollution 172 events at SDZ always coincide with HFP pollution events. The new Fig. S12 shows the ratios of the area 173 response in ambient air samples relative to the working standard (RL (reported)) for c-C4F8 (PFC-318), HFP, 174 and HFC-23. Good correlations among the three compounds are evident. We removed references to other 175 compounds from the text for brevity. We changed the wording from "associated with" to "coincide with" and 176 added a reference to the new Fig. S12 which clearly shows the correlations. As requested we clarified the 177 second sentence to "virtually absent (≤0.01 ppt)".
- 178
- L639-641 Consistent with emissions from many of these facilities, but clearly not all (as stated in L610-612). Given
  the problems with associating these sources can the authors confirm that the ratios between m/z 131 and 101 during
- pollution events were consistent with those observed in clean air? This would help to rule out interferences during
  pollution events.
- We can confirm that even during the highest pollution events measured at Gosan, the ratios of the mass over charge ratios m/z 131 over m/z 100 show no deviation from those observed in bracketing standards or during background conditions.
- 186

187 *L661 How much smaller?*

188 We thank the reviewer for pointing out this omission at the end of Section 5.3.2, which made us also realize

189 that we had not defined the list of countries for the North Western European emissions given in this Section.

- 190 We have now added this information and we also reran the European inversion as a mistake had been found.
  - 5

- 191 This leads to a slight upward revision of the emissions (from  $0.02 \pm 0.01$  Gg yr-1 to  $0.026 \pm 0.013$  Gg yr-1) and
- 192 an updated Fig. 8. We adjusted the wording in Section 5.3.2 to reflect the updated results. We have also added
- 193 that "The inversion is broadly consistent with emissions from PTFE/FEP production and FABS, but emissions
- 194 from other industrial sources may also play a role". As requested, we have now added UNFCCC and bottom-
- up emissions for comparison: 0.0007 Gg yr-1 (UNFCCC, 2013–2014) and 0.0017 Gg yr-1 (Bottom-up emission
- 196 inventories, Section 3, 2013–2014)) for the inversion domain.
- 197
- 198 *L*697 *That is a very optimistic way of looking at that Figure.*

199 The sensitivity of the emissions generally decreases with distance from the measurement location, which leads to increased uncertainty in the inversion, both in the spatial distribution of emissions and their overall 200 201 magnitude. The further away emissions occur, the more likely the regional inversion method will allocate 202 these emissions to a general diffuse region, rather than identify individual c-C4F8 point sources. We added this 203 explanation to Section 4.4 to point out more clearly the limitations of the regional inversion method. Due to 204 the limited number of samples taken onboard the aircraft, the regional inversion for the Indian subcontinent 205 may have more difficulty identifying individual point sources (which also may not be emitting at all times). 206 We added this information in Section 4.6. We modified the sentence in Section 5.3.5. to stress these limitations 207 "Given the limitations of the inversion method to identify distant point sources from a relatively small number of samples (see Sections 4.4 and 4.6), the posterior emissions ...". As pointed out in the text "Emissions 208 209 predominantly occur outside of the Indo-Gangetic plain, the most densely populated region of India" and we now 210 add ", which excludes potential sources that scale with population. Instead the inversion allocates emissions in a much less densely populated region in which multiple likely industrial point sources for c-C4F8 are located." 211 212 We hope that these additional explanations address the reviewer's concern with respect to Fig. 9 and Section 213 5.3.5.

214

**215 L706-707 This is not very clear from the Figure, which is rather indicating an unknown source.**

216 Given that all the potential PTFE/FEP producing facilities we found in India are located within the emissive 217 region identified by the inversion (Fig. 9), while none are in the heavily populated Indo-Gangetic Plain, and 218 keeping the limitations of the inversion in mind (see our reply above), we are confident that the inversion results support our hypothesis that production of PTFE/FEP and other fluorochemicals is the likely dominant 219 220 source of  $c-C_4F_8$  emissions. Still, we modified the sentence to "While we cannot categorically exclude an 221 unknown industrial source, these results are consistent with the chemistry of PTFE/FEP production as dominant emission source of c-C4F8.". We also modified two sentences in the Summary and conclusions section 222 223 accordingly following the reviewer's advice. Based on new information, we also added "Note, that one of the 224 facilities in western India (Navin Fluorine International, Surat, Gujarat) is known to also produce HFO-225 1234yf since 2016, using a process which starts out with the same chemistry, that is the pyrolysis of HCFC-22 to TFE and HFP, with  $c-C_4F_8$  as potential by-product (see Supplement)". We added a short section in the 226 Supplement with citations about HFO-1234yf. We also added a similar sentence in Section 5.3.1 about the 227 228 PTFE production facilities of the Juhua Group Corporation in Zhejiang province which also produce HFO-

1234yf since 2016. Other facilities licensed by Honeywell to produce HFO-1234yf using this route with potential  $c-C_4F_8$  emissions may exist in East Asia, but any such production is relatively recent and cannot explain historic  $c-C_4F_8$  emissions.

232

**233 L713 What is the main purpose of this direct $c-C_4F_8$ production?**

234 The main purpose  $c-C_4F_8$  produced is unfortunately not listed on the HaloPolymer website. We contacted the 235 company, but received no reply. The website broadly states that "R318C is used in air-conditioners, heat 236 pumps and energy units. It is also used for synthesis of fluororganic compounds". The website lists  $CF_4$ ,  $C_2F_6$ , 237 c-C4F8, SF6, and WF6 as "specialty gases (that) are organic and inorganic fluorinated gases widely used as 238 dielectrics and fire extinguishing agents, in dry etching processes during production of microelectronics.", but 239 does not specify for which of these applications exactly  $c-C_4F_8$  is used. The only use that was not included in 240 the Introduction is "for synthesis of fluororganic compounds". When searching for chemical reactions on 241 scifinder.cas.org, several reactions can be found in which  $c-C_4F_8$  is used to introduce -CF3 group into larger 242 organic molecules. The reaction of  $c-C_4F_8$  with TFE to HFP, as described in the Introduction, is also found. 243 Three other reactions are described which lead to a variety of products, including desired products such as 244 the hydrofluoroolefin HFO-1234vf, a fourth generation refrigerant used in newer mobile air conditioners 245 (MACs) or HFP, but also various other products. While it is not clear which of these, or other reactions, using 246  $c-C_4F_8$  as feedstock are commercially important, we added this new information to the Introduction. We have 247 also added short discussions on possible  $c-C_4F_8$  emissions from HFO-1234yf production in recent years.

248

**249 L729 Which ones did PFC-318 correlate best with (also for other pollution events in Asia etc)?**

The best correlation was observed between PFC-318 (c-C4F8) and HFC-23 at ZEP. Other compounds, such as HCFC-22, CFC-13, CH2Cl2, CHCl3, or TCE showed weaker correlations. We added Fig. S15 to show PFC-318 (c-C4F8), HFC-23, and HCFC-22 concentrations at ZEP and now refer to this figure. We also added Fig. S12 which shows the good correlations between PFC-318 (c-C4F8), HFP, and HFC-23 enhancements at SDZ.

254

255 L740-741 How much larger?

We have modified the sentence and added this information in the main text: "These global emissions are significantly larger than what can be compiled from available bottom-up inventory information ( $70 \pm 17$  times, 1990–1996,  $29 \pm 5$  times, 1997–2010,  $15 \pm 1$  times, 2011–2014)".

259

L1236 Figure 1 is mostly demonstrating quality assurance purposes and one cannot see most station data anyway as
it is on top of each other. As the long-term trend is shown again in Figures 3 and 4 I suggest moving it to the
supplement.

We believe that demonstrating data quality assurance and showing the underlying raw data is very important.
 Moreover, as requested we have added the confidence bands and the interhemispheric gradients to Fig. 1 and

265 therefore would like to retain Fig. 1 in the main text.

L1256 Is it necessary to show years from 1900 if the first data point is after 1930? What is the uncertainty of the calculated effective ages?

As  $c-C_4F_8$  mixing ratios are not much different from zero in the early decades of the 1900s, we have changed Fig. 3 to show mixing ratios reconstructed by the CSIRO inversion from 1930. We would like to point out though that firn measurements are not associated with discrete age values, rather they relate to atmospheric mole fraction from a range of times in the atmosphere. The oldest data point is from South Pole and has a mean age of 1890 (it is mentioned in the Fig. 3 caption that it is not plotted). The data point with mean age 1933 reflects a mix of air from about 1900–1950. So although the measurements do contain information about mole fraction before the 1930s, there is not much change occurring in mole fraction or emissions.

276 Effective ages before about 1965 are very uncertain, as they depend on the growth rate of  $c-C_4F_8$  in the 277 atmosphere which itself is quite uncertain and small at this time. However, as described at the end of the 278 second paragraph in Section 4.3.1, mean ages are shown in Fig. 3 (and the new Fig. S7) for firn data that 279 would have an age before 1965 (for the best case estimate). Effective ages after 1965 are also dependent on the 280 atmospheric growth rate, but this is known quite well from the inversion. Note that the firn data are shown 281 versus mean or effective age in Fig. 3 and new Fig. S7 for illustrative purposes only; the CSIRO inversion uses 282 Green's functions (also called age distributions) from the firn model to characterize the age of the air in each 283 firn sample, with the ensemble of Green's functions used to incorporate uncertainty (as described in Section 284 4.3.1). Therefore uncertainty in effective age is only relevant for the comparison in Fig. 3 and new Fig. S7 and 285 not for the CSIRO inversion. The 2 sigma range for effective ages varies between about  $\pm 0.2$  and  $\pm 4$  ppt, with 286 a mean value of ±1.4 ppt (see new. Fig S7).

287

288 Supplement

289 Figure S1 The caption is actually an entire section and should perhaps have its own heading.

We now give the Section its own heading "Details on the tuning of the CSIRO firn model for the Summit13 site" and moved the heading "Supplemental Figures" and Fig. S1 just below this Section.

292

Figure S3 There is quite some uncertainty in the 1960s and 70s. Has this been reflected in the emission uncertainties?

Yes, uncertainty in the diffusion coefficient for  $c-C_4F_8$  relative to  $CO_2$  is included in the Green's function ensemble that is used in the bootstrap method to calculate uncertainties in emissions inferred by the CSIRO

inversion. To clarify this, we modified the last sentence of the first paragraph of Section 4.3.1 to "... different

firn model parameters including relative diffusivity (Trudinger et al., 2013, ...)".

299

300 Figure S7 Please also show the published observational data set of Saito et al.

301 Note, old Fig. S7 is now new Fig. S8. As requested, we added the baseline trends given in Saito et al., 2010.

302 However, we would like to point out that these data had calibration drift problems, see our reply to L547

303 below, which is why we did not include them in the inversion. We also believe that the baseline algorithm used

304 for these data did not work as well as the AGAGE baseline algorithm, perhaps exacerbated by the significant

- pollution observed and worse precisions. Therefore, the seemingly large differences between the Saito et al.
   trend lines themselves and the AGAGE and Oram et al. data are misleading. Note, we also added citations for
   Saito et al. and Oram et al. under new Fig. S8.
- 308
- 309 *Table S3 RoW is not explained and web pages should be cited with the date on which they were accessed.*
- 310 We thank the reviewer for this comment as it made us revisit Tables S3 and S4. We realized that we made
- 311 mistakes with the references and corrected those. We now include the names of the two market reports and
- 312 when each of the three sources was accessed. We also include definitions of RoW (Rest of the World). For the
- 313 two market research reports, RoW includes the market share for India and Russia. For www.qianzhan.com,
- 314 North America and Europe are also included in RoW as they could not be separated.
- 315

- 316 **Reviewer 2:**
- 317
- 318 The manuscript entitled "Perfluorocyclobutane (PFC-318,  $c-C_4F_8$ ) in the global atmosphere" by Mühle et al. has
- 319 been evaluated by this reviewer. The paper presents a substantial piece of measurement and modeling work on the
- 320 atmospheric abundance and emission rates of perfluorobutane. The authors have developed an independent
- 321 gravimetric  $c-C_4F_8$  calibration scale and characterized the abundance of  $c-C_4F_8$  with high precision in both
- 322 hemispheres in order to determine historical emissions (archived samples) and recent global emissions. Using
- 323 inversion modeling techniques, regional emission patterns (and pollution events) are investigated in detail, revealing
- 324 that major  $c-C_4F_8$  sources are found in heavily industrialized provinces of China (and perhaps Russia), due to the
- 325 production of PTFE and other fluoropolymers. They predict  $c-C_4F_8$  emissions will continue to rise and that  $c-C_4F_8$
- 326 will become the second most important PFC emitted to the atmosphere in terms of CO2 equivalent emissions.
- General Comments: The manuscript is a pleasure to read, has very few technical errors, and presents and an impressive amount of interesting data. The authors have done a commendable job to present a succinct and encompassing description of the methods and the results. The conclusions follow elegantly from the data presented and form a compelling narrative, especially considering the magnitude of difference in the potential emissions sources involved. I have only few scientific comments/questions. Those are listed below here, followed by technical
- 332 (suggested) corrections:
- We thank the reviewer for the very positive overall evaluation of our research article. We are very pleased that the reviewer agrees with our line of reasoning and conclusions.
- 335
- 336 Specific Comments:
- 337 L55: ": : : : explaining the increase in emissions." Presumably the authors here refer to the early/pre 1980's?
- 338 This indeed needed a clarification. We added "in the 1960s/70s" in the Abstract.
- 339

340 *L65: "Significant emissions" must be inferred significant emissions?*

- 341 Yes, this is correct as in the difference between global emissions and the sum of regional emissions. We 342 changed as suggested by the reviewer to "significant emissions inferred" in the Abstract.
- 343

344 L115: What is "aerolyzed foods"? Please explain (very briefly), or used more common term.

345 We thank the reviewer for pointing out this mistake in the Introduction. We meant aerosolyzed foods which

346 refers to foamed food products and sprayed food products, but this is perhaps not very commonly used.

- 347 **Therefore, we have replaced this with "foamed/sprayed** foods".
- 348
- 349 L262: Please explain what is meant by "above bubble close-off".

350 Throughout the firn (compacted snow), air is contained in tiny channels that are open to the atmosphere. As

- 351 more snow accumulates at the surface, the weight of the snow above causes the channels to be compressed and
- 352 they eventually close to form discrete bubbles of air embedded in ice. Below this point the air cannot be
- 353 pumped out anymore. We have modified the sentence in third paragraph in Section 2.3 to "... from 19 depth

354 levels in the firn from the surface to 80.06 m (below this depth firn air can no longer be collected as the open 355 channels in the firn have closed off and formed discrete air bubbles embedded in ice)."

356

357 L396-397: "it was assumed that emissions were constant from year to year". This seems confusing to me. Perhaps

358 I'm not understanding this inversion correctly. I can see that the emissions would be assumed constant during the 359 year, but why from year to year? How does this work?

360 We agree that this sentence at the beginning of Section 4.3.2 could be better worded. When a Bayesian 361 inversion is performed, certain "a priori" assumptions need to be made to inform the inversion. These are 362 often times emissions from a bottom-up inventory, which are believed to be reasonably close to reality, but 363 bottom-up emissions for  $c-C_4F_8$  are significantly too low. Therefore our approach, which has been used 364 extensively in the literature, was to assume that emissions in any given year are similar to the previous and the 365 next year, but to allow for a certain change (year-to-year emissions growth), that is we expect emissions to only change gradually. We rephrased the sentence to "A priori, it was assumed that emissions were similar from 366 year to year such that the *a priori* year-to-year emissions growth rate was assumed to be zero with an uncertainty 367 of 200 t yr-2 (0.2 Gg yr-2  $1\sigma$ ), approximately twice the bottom-up estimate in Sect. 3.". Note, that we also corrected 368 the unit from t yr-1 to t yr-2 (Gg yr-1 to Gg yr-2), as it is an uncertainty in the emissions growth rate, and 369 370 specified that it is a  $1\sigma$  uncertainty.

371

L440: "We do not report emission estimates outside of eastern Asia due to large posterior uncertainties but include
them assisted with determination of the boundary conditions". I do not understand this approach. Please clarify and

374 explain.

375 We agree that this sentence in the third paragraph of Section 4.4 was rather confusing. Emission estimates far 376 from the measurement station will be highly uncertain, both in terms of their spatial distribution and 377 magnitude. We therefore choose to only report emissions for a region where the uncertainty is small enough 378 that we are able to draw conclusions from the estimates, here eastern Asia. Nevertheless, the emissions outside 379 of the reported region are still estimated in the inversion as they may contribute to pollution events measured 380 at GSN. The contribution to the absolute error to the modelled mole fraction from distant emissions sources is 381 small, but the resultant uncertainty in their inferred emissions is large. This leads to larger uncertainty in the 382 reported regional emissions if they are included, which may hinder interpretation of results. To clarify, we 383 changed to sentence to "While we do not report emission estimates outside of eastern Asia due to large posterior 384 uncertainties, they are still estimated in the inversion as they are useful when modelling emissions in eastern 385 Asia and their uncertainties that we do report.".

386

387 L538: Please explain why not incorporating the firn data has this impact on the emissions estimates.

388 Prior to 1980, the Bristol inversion is based on sparse, uncertain NH archive data, and the CSIRO inversion

389 on the same NH archive data plus firn data with age distributions covering roughly 40 years. The differences

390 between the inversions before the early 1980s are within the estimated uncertainties for these reconstructions

391 as can be seen in Fig. 5. We modified the sentence at the beginning of Section 5.2 to "The Bristol inversion

initially reconstructs lower emissions, but the differences are within the estimated uncertainties for the
 reconstructed histories (see Fig. 5).".

- 394
- 395

**396 L547: How can the mole fractions of this very unreactive compound change in the tanks?**

397 The ratio of NIES/AGAGE  $c-C_4F_8$  calibration assignments for two tanks exchanged between NIES and AGAGE (SIO) changed by more than 10% between 2008 and 2016, which is completely unacceptable. On the 398 399 contrary, the IN/OUT values assigned by AGAGE (at the beginning and end of each tank's service time), 400 agree for both tanks within precisions of 0.02 ppt (~1.1 to 1.7%). Therefore, we concluded that there must 401 have been an internal calibration drift problem at NIES for  $c-C_4F_8$  in tanks NIES used to assign calibrations 402 to the two tanks exchanged with AGAGE/SIO. Unfortunately, we do not have enough data to characterize this 403 further. One possible explanation for the drift (change of concentration) of such an inert perfluorinated 404 compound could be the presence of Christo-Lube MCG111, which had been used by the manufacturer on a 405 limited number of Essex tanks to deal with leak problems at the valve flange. MCG111 is a mixture of 406 perfluorinated polyether (PFPE) and polyetrafluoroethylene (PTFE). We showed, without a doubt, that it is 407 able to produce polyfluorinated compounds including  $c-C_4F_8$  and  $CF_4$  at ppt level, which caused a good deal of grieve for the AGAGE network. We still think that the sentence in the manuscript describes what the likely 408 409 cause is, even though we did not include any of the details for brevity sake as we did not use the data. We 410 slightly modified this and the previous sentence in the second paragraph of Section 5.2.

411

**412 L553: $C_2F_6$ is here listed as a minor PFC, however in L122, it was a major. Which are the majors and the minors?**

413 We agree that this is not consistent. In terms of mixing ratios, there is only one major PFC, CF4, currently at 414 ~86 ppt in the Northern Hemisphere, while the other three PFCs could all be called minor PFCs with  $C_2F_6$  at 415 ~4.9 ppt,  $c-C_4F_8$  at ~1.8 ppt, and  $C_3F_8$  at ~0.69 ppt. In terms of GWP100 CO2 equivalent emissions, see old Fig. 416 S8/new Fig. S9, CF4 is also in its own league, while C2F6 and c-C4F8 CO2-eq. emissions are similar but smaller 417 and  $C_3F_8$  CO2-eq. emissions are even smaller. Therefore, we modified the sentence on L122 (Introduction) to 418 "While the major atmospheric **PFC**, tetrafluoromethane ( $CF_4$ ) as well as the minor **PFCs** hexafluoroethane ( $C_2F_6$ ) 419 and octafluoropropane ( $C_3F_8$ ) are ...". The use of "minor" in the two statements is now consistent with each 420 other (last sentence of Section 5.2 and seventh paragraph of the Introduction).

421

422 L558: To make clear what we are talking about, I suggest inserting "from a climate forcing standpoint" before "will
423 become the second most important PFC: ::".

424 We chose to modify the sentence at the end of Section 5.2 to: "c-C4F8 CO2-eq. emissions have been ..., so that c-425 C4F8 will become the second most important PFC emitted into the global atmosphere in terms of CO2-eq. 426 emissions."

427

428 "Technical" Comments:

429 L42-43: The propagated uncertainties on the emissions should be given in the abstract.

| 430 | Agreed. We added $1\sigma$ uncertainties to the Abstract: " the 1960s to $1.2 \pm 0.1$ (1 $\sigma$ ) Gg yr -1 in the late 1970s to |
|-----|-----------------------------------------------------------------------------------------------------------------------------------------------|
| 431 | late 1980s, then declined to $0.77 \pm 0.03$ Gg yr -1 in the mid-1990s to early 2000s, rise since the early 2000s to 2.20          |
| 432 | ± 0.05 Gg yr -1 in 2017". We changed this accordingly in the "Summary and Conclusions" section.                                    |
| 433 |                                                                                                                                               |
| 434 | L61: " in agreement with our analysis". This seems like an obvious statement. Suggest deleting.                                               |
| 435 | Agreed. We deleted it from the Abstract. We also deleted this in the "Summary and Conclusions" section.                                       |
| 436 |                                                                                                                                               |
| 437 | L102: "Recently there is also further evidence: : :". This sentence begins awkwardly – suggest rewording it.                                  |
| 438 | We changed the sentence in the Introduction to "Today we also have further evidence that".                                                    |
| 439 |                                                                                                                                               |
| 440 | L249: "Fig.1." Other places in the manuscript "Figure" is used. Check for consistency.                                                        |
| 441 | Our understanding is that ACP requires the use of "Figure" when it stands at the beginning of a sentence and                                  |
| 442 | "Fig." when it stands anywhere else in a sentence.                                                                                            |
| 443 |                                                                                                                                               |
| 444 | L302: Perhaps "default" is a better word in place of "definition".                                                                            |
| 445 | We agree that "by definition" was not the right choice of words, but "by default" does not seem right either.                                 |
| 446 | We modified the sentence in Section 3 to "However, these data are inherently not representative of total global                               |
| 447 | emissions since developing countries do not".                                                                                                 |
| 448 |                                                                                                                                               |
| 449 | L311: Move the definition of $1t = 0001 Gg$ to the introduction paragraph, where $Gg$ is first used.                                          |
| 450 | Done. We moved this to the Introduction.                                                                                                      |
| 451 |                                                                                                                                               |
| 452 | L316: replace "similar to" with "analogous to what has been observed for".                                                                    |
| 453 | Done. We also modified the beginning of the sentence in Section 3: "As has been found by Saito et al. (2010) and                              |
| 454 | Oram et al. (2012), we show in Sect. 5.2 and 5.3 that measurement based ("top-down") global and most regional                                 |
| 455 | emissions are significantly larger than the compiled bottom-up $c-C_4F_8$ emissions inventory information (see Fig. 5),                       |
| 456 | analogous to what has been found for other PFCs (Mühle et al., 2010), reflecting the shortcomings of current                                  |
| 457 | emission reporting requirements and inventories".                                                                                             |
| 458 |                                                                                                                                               |
| 459 | L641: Insert "occurring" before ": : :in China: : :)                                                                                          |
| 460 | Done (last sentence in Section 5.3.1).                                                                                                        |
| 461 |                                                                                                                                               |
| 462 | Line 689-691: These two sentences belong more appropriately in sections 5.35 and 5.3.6.                                                       |
| 463 | On one hand, we agree with the reviewer. On the other hand, these two sentences at the end of Section 5.3.4                                   |
| 464 | serve as transition from 5.3.4 to 5.3.5 and 5.3.6. Moreover, the first sentence applies to both 5.3.5 and 5.3.6 and                           |
| 465 | would have to be repeated if moved. Unless the reviewer feels strongly about this, we prefer to leave it as is.                               |
| 466 |                                                                                                                                               |
| 467 | Figures:                                                                                                                                      |
|     |                                                                                                                                               |

- 468 Figures 1 is nicely formatted, but the formatting is inconsistent with that applied in Figures 2-4. Moreover Figure 5
  469 has a completely different formatting style. This figure formatting ought to be "harmonized".
- 470 For Fig. 1, we increased the fonts sizes and stroke of the box and tick marks (it was also updated following
- 471 reviewer #1's request to add interhemispheric gradient and confidence bands). For Fig. 5, we adjusted the
- 472 fonts, changed the color of the box and tick marks, added an axis at the right and top, and removed the outer
- 473 box. Both Figures now more closely resemble Figures 2 4. We hope that no further changes will be needed.
- 474
- 475 *Figure 1 caption: What are the error bars?*
- 476 We added to the figure caption "For individual samples, error bars reflect measurement precisions. For
- 477 monthly means, error bars represent standard deviations."

**Perfluorocyclobutane $(PFC-318, c-C_4F_8)$ in the global 1**

**atmosphere 2**

3

4

5

Jens Mühle1, Cathy M. Trudinger2, Luke M. Western3, Matthew Rigby3, Martin K. Vollmer4, Sunyoung Park5, Alistair J. Manning6, Daniel Say3, Anita Ganesan7, L. Paul Steele2, Diane J. Ivy8, Tim Arnold9,10, Shanlan Li5, Andreas Stohl11, Christina M. Harth1, Peter K. Salameh1, Archie McCulloch3, Simon O'Doherty3, Mi-Kyung Park5, Chun Ok Jo5, Dickon Young3, Kieran M. Stanley3, Paul B. Krummel2, Blagoj Mitrevski2, Ove Hermansen11, Chris Lunder11, Nikolaos Evangeliou11, Bo Yao12, Jooil Kim1, Benjamin Hmiel13, Christo Buizert14, Vasilii V. Petrenko13, Jgor Arduini15,16, Michela Maione15,16, David M. Etheridge2, Eleni Michalopoulou3, Mike Czerniak17, Jeffrey P. Severinghaus1, Stefan Reimann4, Peter G. Simmonds3, Paul J. Fraser2, Ronald G. Prinn8 and Pav F. Weise1 6

7

8

9

- 10
- Ronald G. Prinn8, and Ray F. Weiss1 11
- 12

13 1Scripps Institution of Oceanography, University of California, San Diego, La Jolla, CA, USA

- 2Climate Science Centre, CSIRO Oceans and Atmosphere, Aspendale, Victoria, Australia 14
- 3School of Chemistry, University of Bristol, Bristol, UK 15
- 4Laboratory for Air Pollution and Environmental Technology, Empa, Swiss Federal Laboratories for Materials 16
- Science and Technology, Dübendorf, Switzerland 17

5KNU, Kyungpook Institute of Oceanography, College of Natural Sciences, Kyungpook National University, South 18 19 Korea

- 20 6Met Office Hadley Centre, Exeter, UK
- 7School of Geographical Sciences, University of Bristol, Bristol, UK 21
- 8Center for Global Change Science, Massachusetts Institute of Technology, Cambridge, MA, USA 22
- 23 9National Physical Laboratory, Teddington, Middlesex, UK
- 10School of GeoSciences, University of Edinburgh, Edinburgh, UK 24
- 11NILU, Norwegian Institute for Air Research, Kjeller, Norway 25

[revised manuscript text omitted]

Comment [JM17]: Rev. #2: L115: What is "aerolyzed foods"? Please explain (very briefly), or used more common term.

**Comment [JM18]:** *Rev. #1: L119 I don't think a personal communication can be counted as evidence*

**Deleted: two**

**Deleted: and**

- ----

**Comment [JM19]:** Rev. #2: L553:  $C_2F_6$  is here listed as a minor PFC, however in L122, it was a major. Which are the majors and the minors?

170 noted that the global emissions determined by Saito et al. (2010) were lower than their estimate and suggested that 171 the underlying atmospheric rise rate measured by Saito et al. may be too small.

172 In summary, calibration differences between previous studies are significant, no multi-decadal c-C4F8 record for the 173 NH has been published, and global emissions have not been reassessed since Oram et al. (2012). Therefore our 174 primary goals have been to develop an independent gravimetric c-C4F8 calibration scale and to characterize the 175 abundances of c-C4F8 with high precisions in both hemispheres in order to determine updated historic and recent 176 global emissions. We present measurements of  $c-C_4F_8$  with precisions of  $\sim 1-2$  % on the SIO-14 calibration scale ( $\sim 2$ 177 % accuracy) developed by the Scripps Institution of Oceanography (SIO) using instrumentation and calibration 178 methods of the Advanced Global Atmospheric Gases Experiment (AGAGE) program (Prinn et al., 2018). We 179 discuss historic atmospheric mole fractions of  $c-C_4F_8$  based on measurements of the CGAA for the extra-tropical SH, 180 archived air samples from various sources for the extra-tropical NH, continuous atmospheric measurements in both 181 hemispheres at multiple remote AGAGE stations since mid-2010, combined with measurements of air extracted from 182 firn from both hemispheres. Using our measurements and inverse modelling methods, we infer global  $c-C_4F_8$ emissions since the beginning of the 20th century until 2017. To improve our understanding of prominent c-C4F8 183 184 sources and source regions, we investigate regional  $c-C_4F_8$  emission strengths as observed by the global AGAGE 185 network in eastern Asia, Europe, parts of Australia and Russia and by an aircraft campaign over India. We also summarize and discuss available inventory based "bottom-up" emissions and compare them to the emissions we 186 187 determined with our atmospheric measurement based "top-down" approach.

**188 2 Experimental methods**

**189 2.1 Instrumentation, data availability, and calibration**

190  $c-C_4F_8$  and ~40 other halogenated compounds were measured by AGAGE in 2 L air samples with the "Medusa" 191 cryogenic pre-concentration system with gas chromatograph (GC, Agilent 6890) and quadrupole mass selective 192 detector (MSD) (Miller et al., 2008; Prinn et al., 2018). Data from twelve in situ measurements sites and fourteen 193 Medusa instruments were used. At Monte Cimone, Italy, c-C4F8 was measured with a commercial Adsorption-194 Desorption System with gas chromatograph and mass spectrometer (ADS-GC/MS) (Maione et al., 2013). Table 1 195 shows the availability of in situ, archived air (Sect. 2.2), firn air (Sect. 2.3) and aircraft air sample (Sect. 2.4) 196 measurements with information for each site. For all measurements, each sample was alternated with a reference gas 197 (Prinn et al., 2000; Miller et al., 2008), resulting in up to 12 fully calibrated samples per day (Medusa and ADS-198 GC/MS). The reference gases at each site were calibrated relative to parent standards at SIO.

199 c-C4F8 measurements are reported on the SIO-14 calibration scale as ppt dry-air mole fractions. The calibration scale

- 200 is based on four gravimetric halocarbon/nitrous oxide (N2O) mixtures via a stepwise dilution technique with large
- dilution factors for each step ( $10^3$  to  $10^5$ ) (Prinn et al., 2000; 2001). High purity c-C4F8 (99.999 %, Matheson Trigas)
- and N2O (99.9997 %, Scott Specialty Gases) were further purified by repeated cycles of freezing (-196° C), vacuum
- 203 removal of non-condensable gases, and thawing. Artificial air (Ultra Zero Grade, Airgas) was further purified via an
- absorbent trap filled with glass beads, Molecular Sieve (MS) 13X, charcoal, MS 5Å, and Carboxen 1000 at -80° C

**Comment [JM110]:** Rev. #1: L169-178 Have the authors ascertained that their calibration system has a linear response behaviour over a relevant mole fraction range? How was the calibration scale uncertainty estimated?

See our reply.

205 (ethanol/dry ice). Zero air was measured to verify insignificant c-C4F8 and other halocarbon blank levels before 206 being spiked with the c-C4F8/N2O mixtures. The resulting mixtures of c-C4F8 in artificial air have prepared values of

[revised manuscript text omitted]

242 limits and within the typical precisions, we chose not to correct for this.

Comment [JM111]: Rev. #1:L187 "perhaps slightly better"? Deleted: , perhaps slightly better for 5975

| λ  | Comment [JM112]: Rev. #1: L210 "perhaps"? |
|----|-------------------------------------------|
| -{ | Deleted: was                              |
| -  | Deleted: ed for                           |
| Υ  | Deleted: perhaps                          |

**247 2.2 Archived air samples of the extra-tropical Southern Hemisphere (SH, Cape Grim Air Archive, CGAA) 248 and extra-tropical Northern Hemisphere (NH)**

249 To reconstruct the atmospheric history of  $c-C_4F_8$  in the extra-tropical SH, 41 unique CGAA samples (collected 250 1978-2009, Langenfelds et al., 2014) were measured at CSIRO in 2011 (Ivy et al., 2012). In addition, 8 subsamples of CGAA parent tanks and four additional SH samples were measured at SIO to demonstrate that measurements at 251 252 CSIRO and SIO agree (for details see the Supplement), Based on an iterative filtering process designed to reject 253 outliers greater than  $2\sigma$  deviations from curve fits through the results for all 60 SH samples (41 at CSIRO and 19 at 254 SIO) and pollution filtered monthly mean measurements (O'Doherty et al., 2001; Cunnold et al., 2002) at the extra-255 tropical stations CGO and ASA (Australia), 13 SH samples were rejected as outliers, leaving 47 SH samples (78 %). 256 To reconstruct the atmospheric history in the extra-tropical NH, 126 unique air samples mostly filled at SIO and THD (1973-2016) were measured at SIO. Additionally, 3 NH samples (filled in 1980 and 1999) were measured at 257 258 CSIRO to demonstrate that measurements at CSIRO and SIO agree (for details see the Supplement). Most of the NH 259 samples had been filled during baseline conditions for various purposes using modified diving compressors (RIX 260 Industries, US, SA-3 and SA-6, Weiss and Keeling laboratories) and did not show any artefacts for many gases (e.g. 261 Mühle et al., 2010; O'Doherty et al., 2014; Vollmer et al., 2016). For c-C4F8, however, comparisons with concurrent 262 in situ measurements at MHD, THD, SIO and JFJ revealed artefacts for most of these samples and the iterative 263 filtering process only retained  $c-C_4F_8$  data for eleven NH samples. In contrast, CGAA tanks were filled with a 264 cryogenic method which did not produce any bias. Due to the sparse NH data and poor data quality before in situ measurements started in the NH, the fits used for the iterative filtering process of NH data had to be guided by the 265 266 final SH fit shifted by 1.5 years to allow for the delay of  $c-C_4F_8$  accumulation between the SH and NH due to interhemispheric transport (Mühle et al., 2010; Vollmer et al., 2016). Without this guidance, initial NH fits were 267 dominated by high outliers, resulting in bad fits. It should be pointed out that most of the filtered NH tanks were 268 269 filled in 2003 and later, typically many tanks on one or two days in a given year, which would add little information 270 to the reconstruction given the onset of in situ data at multiple stations in 2011 and the high quality of the CGAA 271 data used to guide the filtering. Fig. 1 shows the filtered data and the final suggested fits and 95% confidence bands.

**272 2.3 Air extracted from firn**

273 To augment the data set of in situ and archived air measurements, we measured c-C4F8 in samples from a subset of

the firn sites described in Trudinger et al. (2016), namely NEEM08 in the NH and DSSW20K and SPO01 in the SH,

275 plus one new site in the NH, Summit13, Greenland. We used the CSIRO firm model (Trudinger et al., 1997;

- 276 Trudinger et al., 2013) to characterize the age of the air in these samples (detailed in Sect. 4.1). Here, we give a brief
- 277 description of the firn sites. For a full description of the calibration of the CSIRO firn model for NEEM08,
- 278 DSSW20K, and SPO01 see Trudinger et al. (2013), and for Summit13 see Fig. S1.
- NEEM08: Firn air was extracted from the EU borehole in July 2008 in northern Greenland, drilled near the North
  Greenland Eemian Ice Drilling Project (NEEM) deep ice core drilling site (77.45° N, 51.06° W) (Buizert et al.,
  2012). This site has a moderate snow accumulation rate of 199 kg m-2 yr-1.
- 282 Summit13: Firn air was collected in May 2013 at Summit, Greenland from a borehole (72.66° N, 38.58° W) drilled
- 283 10 km NNW of Summit Station, Greenland. The US Firn Air system (Battle et al., 1996) was used to extract the air

**Comment [JM113]:** Rev. #1: L214-249 It is commendable that the authors have carried out these tests. However, the high number of statistical outliers is worrying and casts some doubt on the derived longterm trends, in particular the early parts. Adding uncertainty ranges to the fits based on a) the samples that were included but showed discrepancies between Medusas and b) the sparsity of the measurements, might help here. In addition, I recommend moving this rather technical paragraph to the supplement.

[revised manuscript text omitted]

Comment [JM114]: Rev. #2: L262: Please explain what is meant by "above bubble close-off". Deleted: bubble close-off (80.06 m

| Deleted: a        |
|-------------------|
| Deleted: [        |
| Deleted: ]        |
| Deleted: [        |
| Deleted: ]        |
| Deleted: (        |
| Deleted: R        |
| Deleted: Gas      |
| Deleted: Detector |

**377 3 Bottom-up emission inventories (UNFCCC, EDGAR, NIRs, WSC)**

378 Emissions of compounds, such as  $c-C_4F_{8}$ , into the atmosphere are often estimated by so called "bottom-up" methods, 379 which are based on information such as purchased, produced or imported amounts, industrial activities referred to as activity data and estimated emissions factors for each emissive process. Developed countries report annual emissions 380 381 of GHG, including c-C4F8, to the UNFCCC using such bottom-up methods. However, these data are inherently not 382 representative of total global emissions since developing countries do not have the same comprehensive UNFCCC 383 reporting requirements, including countries such as South Korea, China, and Taiwan with sizable electronics and PTFE manufacturing capacities and thus with potentially significant  $c-C_4F_8$  emissions. An additional complication is 384 385 that several countries report unspecified mixes of PFCs or of PFCs and HFCs and other fluorinated compounds, 386 making it difficult or impossible to estimate emissions of individual compounds, such as  $c-C_4F_8$ . In the Supplement, 387 we gather available inventory information from submissions to UNFCCC, National Inventory Reports (NIRs), the 388 Emissions Database for Global Atmospheric Research (EDGAR), the World Semiconductor Council (WSC), and the 389 US Environmental Protection Agency (EPA) in an effort to estimate contributions from unspecified mixes and 390 countries not reporting to UNFCCC to compile a meaningful bottom-up inventory. Globally these add up to 10-30 t yr-1 (0.01–0.03 Gg yr-1) from 1990 to 1999, 30–40 t yr-1 (0.03–0.04 Gg yr-1) from 2000 to 2010, and 100–116 t yr-1 391 392 (~0.1 Gg yr-1) from 2011 to 2014 (with a substantial fraction due to the U.S. emissions from fluorocarbon production reported by US EPA). As has been found by Saito et al. (2010) and Oram et al. (2012), we show in Sect. 5.2 and 5.3 393 394 that measurement based ("top-down") global and most regional emissions are significantly larger than the compiled bottom-up  $c-C_4F_8$  emissions inventory information (see Fig. 5), analogous to what has been found for other PFCs 395 396 (Mühle et al., 2010), reflecting the shortcomings of current emission reporting requirements and inventories.

**397 4 Modelling studies**

**398 4.1 CSIRO firn model**

399 The CSIRO firn model and its use in global inversion frameworks has been described in detail (Trudinger et al., 400 2013; Trudinger et al., 2016; Vollmer et al., 2016; Vollmer et al., 2018; Vollmer et al., 2019). Air samples taken far 401 away from pollution sources represent the background atmospheric trace gas composition at that time. Once air enters the firn, vertical diffusion and other physical processes in the firn lead to mixing of air of different ages. 402 403 Therefore, air extracted from firn must be described with an age distribution. We used the CSIRO firn model to 404 describe the relationship between trace gas mole fractions measured in each extracted air sample from a given depth 405 and the corresponding age distribution of high-latitude atmospheric mole fractions. The diffusion coefficient of c-406  $C_4F_8$  relative to that of  $CO_2$  in air at 253 K used here was 0.47 with an estimated uncertainty of ~10 %. This value was determined using Equation 4 from Fuller et al. (1966) with Le Bas volume increments (e.g. Table 1.3.1, Mackay 407 et al. (2006) and a multiplier for the Le Bas increments of 0.97 (which minimizes the difference of calculated relative 408 409 diffusion coefficients of a number of compounds from values measured by Matsunaga et al. (1993, 2002, 2005)).

| Comment [JM115]: Rev. #2: L302: Perhaps
" default " is a better word in place of "definition ". |  |  |
|---------------------------------------------------------------------------------------------------------------------------------------|--|--|
| Deleted: and ozone depleting substances (ODS)                                                                                  |  |  |
| Deleted: T                                                                                                                            |  |  |
| Deleted: however, by definition,                                                                                                      |  |  |
| Deleted: as                                                                                                                           |  |  |

| Comment [JM116]: Rev. #2: L311: Move the definition of 1t =0001Gg to the introduction paragraph, where Gg is first used. |
|---------------------------------------------------------------------------------------------------------------------------------|
| Deleted: , 1 t = 0.001 Gg                                                                                                |
| Deleted: Similar to what                                                                                                        |
| Deleted: d                                                                                                                      |
| Deleted: pointed out                                                                                                            |
| Deleted: will                                                                                                                   |
| Comment [JM117]: Rev. #2: L316: replace
"similar to" with "analogous to what has been
observed for".               |
| Deleted: similar                                                                                                                |
| Deleted: to                                                                                                                     |
| Deleted: emission                                                                                                               |
|                                                                                                                                 |

[revised manuscript text omitted]

Comment [JM120]: Rev #1. L513 How high is the interhemispheric gradient and how has it evolved over time? This might e.g. reveal information on changes in emission latitudes. There is a lot of space in Figures 1 and 4 to show this.

Comment [JM121]: Rev. #1: L523 Define "good agreement". There are no uncertainty ranges given for the two estimates in Figure S7.

Now shown in new Fig. S7. Deleted: down

| Deleted: S7                                                                                                                                                      |
|------------------------------------------------------------------------------------------------------------------------------------------------------------------|
|                                                                                                                                                                  |
| Deleted: ,                                                                                                                                                       |
| Deleted: S7                                                                                                                                                      |
|                                                                                                                                                                  |
| Comment [JM122]: Rev. #2: L538: Please
explain why not incorporating the firn data has this
impact on the emissions estimates. |
| Comment [JM123]: Rev. #1: L537-538 Again, are these discrepancies within uncertainties?                                                            |
| Deleted: catches up by the early 1980s, perhaps because firn data were not incorporated.                                                                  |

emissions determined by both inversions declined to  $\sim 0.8$  Gg yr-1 in the mid-1990s to early 2000s. Since then emissions kept increasing, reaching  $\sim 2.2$  Gg yr-1 in 2017. Both inversions reconstruct emissions which are significantly larger than available bottom-up inventory information (see Sect. 3 and the Supplement), reflecting the shortcomings of the current UNFCCC reporting requirements and bottom-up inventories.

Emissions presented by Oram et al. (2012) agree very well from 2001 to 2007 with our results and on average also from 1978 to 2001, although they show larger variability. Global emissions roughly estimated by Harnisch (2000)

based on measurements by Travnicek (1998) of ~0.7 Gg yr-1 from 1978 to 1997 are 30% lower than our estimate of 1.01  $\pm$  0.10 Gg yr-1. Saito et al. (2010) estimated global emissions of 0.6  $\pm$  0.2 Gg yr-1 from January 2006 to

673 September 2009, about half of our  $1.16 \pm 0.09$  Gg yr-1 estimate. This difference is likely due to slowly changing *c*-

significantly affect the background rise rate and thus global emissions, but would have had less influence on the

regional emissions estimated by Saito et al. (2010) as these are mostly dependent on the magnitude of the much

Global emissions of c-C4F8 have clearly not levelled off at 2005–2008 levels as had been suggested by Oram et al. (2012), but kept rising. In contrast, emissions of other minor PFCs, C2F6 and C3F8, have decreased since the early

2000s and stabilized in recent years (Trudinger et al., 2016), reflecting that emission sources and/or use patterns of c-

 $C_4F_8$  are different from those of the other minor PFCs. Weighted by GWP100 (100-year timescale) estimated 2017

emissions of c-C4F8, C3F8, C2F6, and CF4 were 0.021, 0.005, 0.022, and 0.083 billion tonnes of CO2-eq., respectively

(see Fig. \$9).  $c-C_4F_8$  CO2-eq. emissions have been larger than those of C3F8 since 2004 and, assuming continued

growth, will also surpas  $C_2F_6$  emissions within a year or two, so that  $c-C_4F_8$  will become the second most important

PFC emitted into the global atmosphere in terms of CO2-eq. emissions. In the next section, we will investigate

regional emissions of c-C4F8 to gain a better understanding how individual regions and sources may contribute to the

 $C_4F_8$  mole fractions in calibration tanks used by NIES (Takuya Saito, personal communication, 2018), which would

**Comment [JM124]:** Rev. #2: L547: How can the mole fractions of this very unreactive compound change in the tanks?

See our reply

**Comment [JM125]:** Rev. #2: L553: C2F6 is here listed as a minor PFC, however in L122, it was a major. Which are the majors and the minors?

Comment [JM126]: Rev. #1: L555 GWP-100?

**Deleted: S8**

**Comment [JM127]:** Rev. #2: L558: To make clear what we are talking about, I suggest inserting "from a climate forcing standpoint" before "will become the second most important PFC: :: "."

688 5.3 Regional *c*-C4F8 emission studies

global emissions.

larger pollution events above background.

**689 5.3.1 Emissions from eastern Asia**

[revised manuscript text omitted]

**Comment [JM128]:** Rev. #1: L571-574 Given that the largest emissions appear to occur near the sea, is there scope for some emissions being related to ships or submissions? What fraction of emissions did the model initially assign to have occurred over the ocean?

**See our reply. Deleted: Compared to t Deleted: , which Deleted: this represents an increase of Deleted: a Deleted: a Deleted: ing Deleted: 1**

**Comment [JM129]:** *Rev. #1: L590-591 This appears to be in disagreement with the statement in L576-578.* See our reply.

Comment [JM130]: Rev. #1: L603 FABS?

[revised manuscript text omitted]

| Deleted: indeed                                                                                                                                                                                                                                                                                                                              |
|----------------------------------------------------------------------------------------------------------------------------------------------------------------------------------------------------------------------------------------------------------------------------------------------------------------------------------------------|
| Deleted: HCFC-22,                                                                                                                                                                                                                                                                                                                            |
| Deleted: CHCl 3 and                                                                                                                                                                                                                                                                                                               |
| Deleted: strong                                                                                                                                                                                                                                                                                                                              |
| Deleted: which are                                                                                                                                                                                                                                                                                                                           |
| Deleted: associated                                                                                                                                                                                                                                                                                                                          |
| Deleted: ,                                                                                                                                                                                                                                                                                                                                   |
| Deleted: CHCl 3 , HCFC-22 and                                                                                                                                                                                                                                                                                                     |
| Deleted: are                                                                                                                                                                                                                                                                                                                                 |
| Deleted: associated                                                                                                                                                                                                                                                                                                                          |
| Deleted: ,                                                                                                                                                                                                                                                                                                                                   |
| Deleted: CHCl 3 , HCFC-22                                                                                                                                                                                                                                                                                                         |
| Deleted: co-occur                                                                                                                                                                                                                                                                                                                            |
| Deleted: are associated with                                                                                                                                                                                                                                                                                                                 |
| Comment [JM131]: Rev. #1: L631-638 Please
add information such as measurement precisions,
observed mole fraction ranges, ions used for
identification and quantification, etc. on the HFP
measurements to the manuscript. Please provide
quantitative evidence instead of "associated with"
and "virtually absent". |
| Comment [JM132]: Rev. #2: L641: Insert
"occurring" before ": : :in China: : :)                                                                                                                                                                                                                                                            |
| Comment [JM133]: Rev. #1 L639-641
Consistent with emissions from many of these
facilities. but clearly not all (as stated in L610-612).                                                                                                                                                                    |

Given the problems with associating these sources

can the authors confirm that the ratios between m/z 131 and 101 during pollution events were consistent

with those observed in clean air? This would help to rule out interferences during pollution events.

[revised manuscript text omitted]
 Comment [JM135]: Rev. 2: Line 689-691: These two sentences belong more appropriately in sections 535 and 536

See our reply

Comment [JM136]: Rev. #1: L697 That is a very optimistic way of looking at that Figure.

| Deleted: Perhaps even clearer than in eastern Asia,                                                                                   |  |  |
|---------------------------------------------------------------------------------------------------------------------------------------|--|--|
| Deleted: point to                                                                                                                     |  |  |
| Comment [JM137]: Rev. #1: L706-707 This is
not very clear from the Figure, which is rather
indicating an unknown source. |  |  |
| Deleted:                                                                                                                              |  |  |
| Deleted: of $c$ -C 4 F 8                                                                                 |  |  |

[revised manuscript text omitted]

| - | Deleted: T                                                                                              |
|---|---------------------------------------------------------------------------------------------------------|
| Ч | Deleted: s                                                                                              |
|   |                                                                                                         |
| - | Comment [JM138]: Rev. #1: L713 What is the main purpose of this direct c-C4F8 production? |
| - | Deleted: S12                                                                                            |
| - | Deleted: PTFE production                                                                                |
|   |                                                                                                         |

**Comment [JM139]:** *Rev. #1:* L729 Which ones did PFC-318 correlate best with (also for other pollution events in Asia etc)?

 $\label{eq:Deleted: the production of } \textbf{Deleted: the production of }$

| - | Deleted: | better            |
|---|----------|-------------------|
| - | Deleted: | versus $\leq$ 7 % |

Comment [JM140]: Rev. #1: L740-741 How much larger?

| istern             |
|--------------------|
| sions,             |
| inces              |
| South              |
| itters.            |
| sions              |
| TFE <mark>/</mark> |
| China              |
| lying              |
|                    |
| 0.20)              |
| utions      |
| tion.              |
| small              |
| ast to             |
| may                |
|                    |
| NW                 |
|                    |
| 1s are             |
| from               |
|                    |
| imate              |
| ld be              |
| n 5 to             |
|                    |
| from               |
| 8 %),              |
|                    |
| s un-              |
| rious              |
|                    |
| annot              |
| may                |
| Pont               |
| by-                |
| ere in             |
| ohere,             |
|                    |
|                    |

Dei

[revised manuscript text omitted]